# SUMO targets the APC/C to regulate transition from metaphase to anaphase

Karolin Eifler[1,4], Sabine A.G. Cuijpers[1], Edwin Willemstein[1], Jonne A. Raaijmakers[2], Dris El Atmioui[3], Huib Ovaa[3], René H. Medema[2] & Alfred C.O. Vertegaal [1]

Signal transduction by small ubiquitin-like modifier (SUMO) regulates a myriad of nuclear processes. Here we report on the role of SUMO in mitosis in human cell lines. Knocking down the SUMO conjugation machinery results in a delay in mitosis and defects in mitotic chromosome separation. Searching for relevant SUMOylated proteins in mitosis, we identify the anaphase-promoting complex/cyclosome (APC/C), a master regulator of metaphase to anaphase transition. The APC4 subunit is the major SUMO target in the complex, containing SUMO acceptor lysines at positions 772 and 798. SUMOylation is crucial for accurate progression of cells through mitosis and increases APC/C ubiquitylation activity toward a subset of its targets, including the newly identified target KIF18B. Combined, our findings demonstrate the importance of SUMO signal transduction for genome integrity during mitotic progression and reveal how SUMO and ubiquitin cooperate to drive mitosis.

[1] Department of Molecular Cell Biology, Leiden University Medical Center, Albinusdreef 2, Leiden 2333 ZA, The Netherlands. [2] Oncode Institute and Division of Cell Biology, The Netherlands Cancer Institute, Plesmanlaan 121, Amsterdam 1066 CX, The Netherlands. [3] Oncode Institute and Department of Chemical Immunology, Leiden University Medical Center, Albinusdreef 2, Leiden 2333 ZA, The Netherlands. [4] Present address: Institute of Molecular Biology gGmbH, Ackermannweg 4, Mainz 55128, Germany. Correspondence and requests for materials should be addressed to K.E. (email: k.eifler-olivi@imb-mainz.de) or to A.C.O.V. (email: vertegaal@lumc.nl)

Faithful copying of the genetic information and accurate separation of chromosomes during mitosis are essential to maintain genomic integrity. Unrepaired DNA damage and unbalanced separation of chromosome pairs in mitosis lead to loss of genomic integrity including aneuploidy and can potentially lead to pathology including cancer[1–3]. Cell cycle progression is exquisitely regulated by protein posttranslational modifications (PTMs) including phosphorylation and ubiquitylation[4]. Enzymes that mediate the conjugation and de-conjugation of PTMs are key drug targets[5].

We are limited in our understanding of the intricate interplay between different PTMs. The complexity of these PTMs at the proteome-wide scale is overwhelming[6]. Kinases play a particularly well-known role in cell cycle progression. The abundance of critical cell cycle components is regulated by the ubiquitin–proteasome system, with a dominant role for the ubiquitin E3 ligase anaphase-promoting complex/cyclosome (APC/C)[7,8].

The APC/C is a 1.2 MDa complex, comprised of 15 subunits, including structural parts like APC1, APC4, and APC5, catalytic components, and the two substrate adapters known as co-activators CDH1 and CDC20[8]. Two different ubiquitin E2s aid the APC/C to ubiquitylate its substrates, UBE2C and UBE2S[9]. The APC/C initiates mitotic exit and governs the progression to G1 phase by targeting key regulators, such as Cyclin B and Securin, for proteasomal degradation[10]. Securin is the inhibitor of the Cohesion cleaving protein Separase. The timely destruction of these regulators is essential for an error-free chromosomal segregation and successful cell division. Therefore, activity of the APC/C is tightly controlled by binding of inhibitors and activators, destabilization of its subunits, and PTMs, such as phosphorylation[10–13].

Deregulation of these control mechanisms and altered activity of the APC/C can therefore lead to severe mitotic defects and genome instabilities and has been associated with the development of various human cancer types[14–18].

In addition to ubiquitin, ubiquitin family members NEDD8 and small ubiquitin-like modifier (SUMO) also contribute to proper cell cycle progression. NEDD8 is a key activator of Cullin-like RING ligases, by modifying a conserved lysine in the Cullin subunits[19]. SUMOs are predominantly conjugated to nuclear proteins and regulate all nuclear processes[20,21]. SUMO conjugation is regulated by a single E2 , UbE2I, previously known as UBC9[22]. Intriguingly, disruption of the UBC9 gene in yeast was found to block cell cycle progression, leading to a block in G2 phase or in early mitosis[23]. Mice lacking UBC9 die at an early post-implantation stage, showing defective chromosome segregation, resulting in anaphase bridges[24].

We are still limited in our understanding of the target proteins regulated by SUMO during cell cycle progression[25]. Here we show that disrupting SUMO signal transduction results in a delay in mitosis and causes defects in mitotic chromosome separation. Searching for relevant SUMOylated proteins in mitosis, we identify the APC/C as a SUMO-regulated target. SUMOylation enhances the activity of the APC/C to a subset of its targets. This work represents a prime example of how SUMO and ubiquitin cooperate to drive mitosis.

## Results

**Inhibition of SUMOylation leads to mitotic delay.** To enhance our insight into the role of SUMOylation[24–26] specifically during mitosis, we have produced HeLa cell lines stably harboring inducible knockdown constructs for both subunits of the SUMO-activating enzyme (SAE1 and SAE2). These cells were analyzed by live cell microscopy to monitor the amount of time needed for full mitotic progression from nuclear envelope breakdown (NEB) until the separation of the sister chromatids in anaphase (Fig. 1a). We have quantified both the amount of time needed from nuclear envelope breakdown until the alignment of the chromosomes at the spindle equator during metaphase as well as the time that passed from metaphase until sister chromatid separation in anaphase for 200 mitotic cells per condition resulting from three independent experiments (Fig. 1b). While the control cells needed on average 12 min to reach the beginning of metaphase, knockdown of SAE1 led to a significant delay of 10 min until proper chromosome alignment. Knockdown of the second subunit SAE2 had a more modest effect on the progression from NEB to metaphase resulting in a delay of about 2–3 min. A much stronger effect of the SAE knockdown was visible on the progression from metaphase to anaphase. While control cells finished chromosomal segregation 35 min after NEB, knockdown of both SAE1 and SAE2 led to a prolonged retention in metaphase. Cells with SAE2 knockdown entered anaphase only about 45 min after NEB and cells expressing the SAE1 knockdown construct were even further delayed showing the onset of anaphase only 65 min after NEB. Knockdown efficiency was verified via immunoblot analysis (Fig. 1c).

We studied mitotic defects by quantifying chromatin bridges between nuclei (Fig. 1d, e). While only 1.2% of the HeLa control cells showed formation of DNA bridges between nuclei, the formation of these chromatin bridges was significantly increased to 5.2% for knockdown of SAE1 and to 10.9% for knockdown with ishSAE2.1. Knockdown of SAE2 with ishSAE2.2 led to a more modest increase in the formation of chromatin bridges between 3.4% of the nuclei, which is in accordance with a milder decrease in SUMO2/3 conjugates after knockdown with ishSAE2.2 compared to ishSAE2.1 (Fig. 1c).

Depletion of SUMO conjugates in the human colon carcinoma cell line HCT116 and the fibrosarcoma cell line HT-1080 by knockdown of either the SAE or the SUMO-conjugating enzyme (UBC9) by infection with lentivirus expressing short hairpin RNAs (shRNAs) against these enzymes had an even bigger effect on the formation of chromatin bridges (Supplementary Fig. 1). While there were no chromatin bridges visible in the cells treated with control shRNA, knockdown of either SAE2 or UBC9 led to a significant increase in the formation of chromatin bridges. Knockdown of SAE2 in HCT116 caused chromatin bridges between 35% of the cells treated with shSAE2.1 and 20% of the cells treated with shSAE2.2. HCT116 cells treated with shRNAs against UBC9 showed chromatin bridges between 36% of the cells in the case of shUBC9.1 and 11% in the case of shUBC9.2. Knockdown of SAE2 by simultaneously treating HCT116 cells with shSAE2.1 and shSAE2.2 did only slightly increase the formation of chromatin bridges to 42%. We obtained similar results when treating HT-1080 cells with shRNAs against SAE2 and UBC9. Knockdown with shSAE2.1 led to the formation of chromatin bridges between 17% of the nuclei and shSAE2.2 increased bridge formation to 11%. Knockdown of UBC9 by shUBC9.1 caused bridge formation between 22% and knockdown of shUBC9.2 between 8% of the cells. The combination of shSAE2.1 and shSAE2.2 did not further increase bridge formation and resulted in DNA bridges between 22% of the nuclei.

Knockdown of SAE2 with ishSAE2.1 had the strongest effect on the reduction of SUMO conjugates (Fig. 1c). This knockdown also caused a severe increase in the formation of DNA bridges, suggesting that the inability to form SUMO conjugates directly correlates with the induction of chromosomal segregation defects (Fig. 1e). On the other hand, knockdown of SAE1 had a lesser effect on the SUMO conjugation levels but still caused the biggest delay in mitosis (Fig. 1b), suggesting that this subunit is specifically important for the cellular signals needed to promote

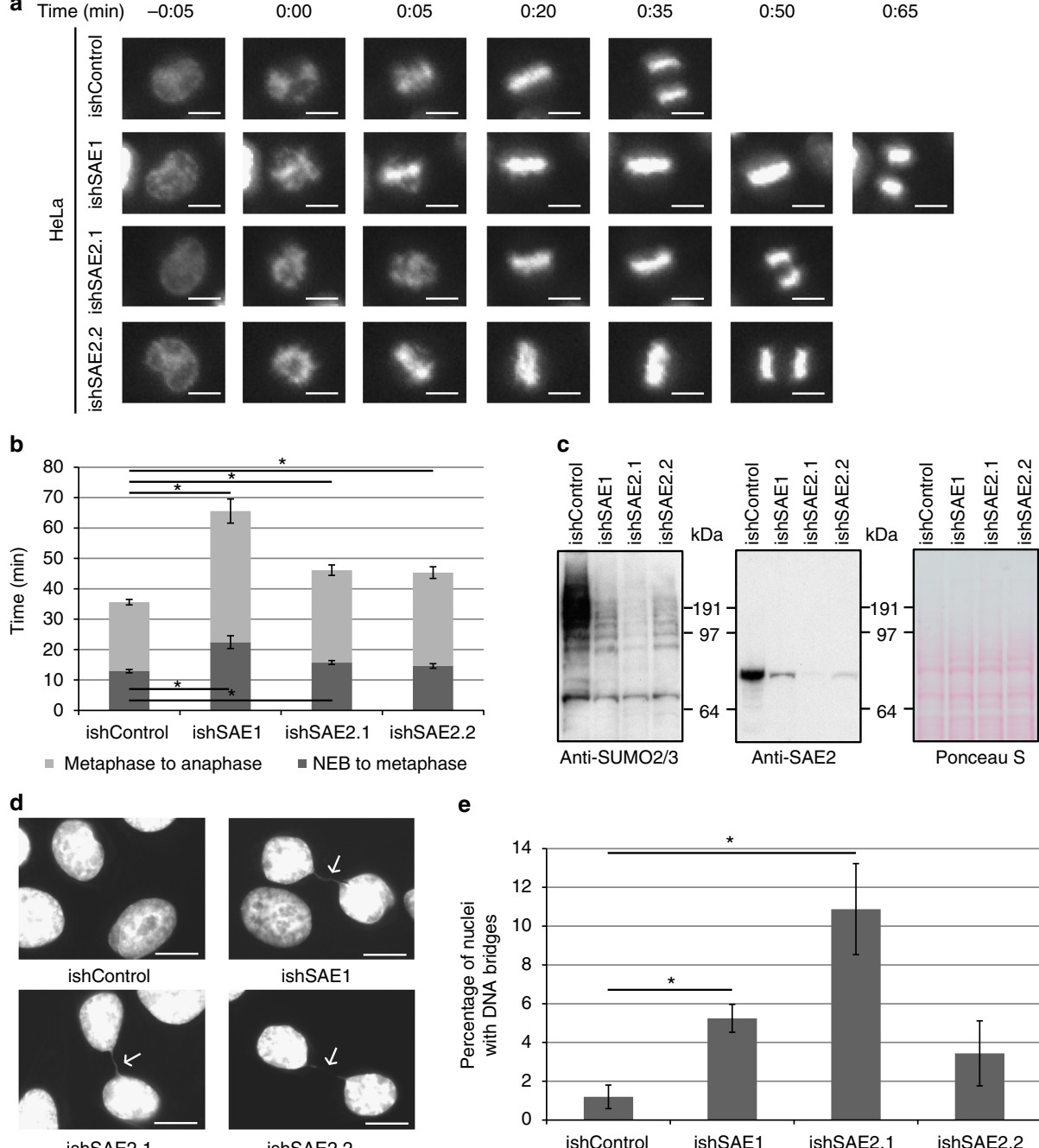

**Fig. 1** Knockdown of the SUMO conjugation pathway delays transition through mitosis and induces the formation of chromosome bridges. **a** Knockdown of the SUMO-activating enzyme (SAE) in HeLa cells was achieved by stably expressing inducible shRNAs generated against both SAE subunits SAE1 (ishSAE1) and SAE2 (ishSAE2.1 and ishSAE2.2). Scrambled shRNA was used as a control (ishControl). These cells were analyzed by live cell microscopy 48 h after induction of the SAE knockdown. Pictures were acquired every 5 min and a selection of pictures is depicted here. Scale bars correspond to 10 μM. **b** The amount of time needed for cells to pass from nuclear envelope breakdown (NEB) to metaphase (dark grey) and from metaphase to anaphase (light grey) was quantified by live cell imaging. Standard deviations were calculated for >200 cells resulting from three independent experiments. A two-sided Student's *t*-test was performed. *$p$-values < 0.05. **c** To confirm a reduction of SUMO conjugates and SAE2 expression, lysates of HeLa cells expressing inducible SAE knockdown constructs were analyzed by immunoblotting with anti-SUMO2/3 and anti-SAE2 antibody 48 h after induction and compared to the control. Equal loading of the lysates was verified by staining with Ponceau S. **d** HeLa cells expressing inducible SAE knockdown constructs were fixed 72 h after induction and stained with Hoechst to monitor the formation of chromosome bridges (white arrows). Scale bars correspond to 10 μM. **e** The percentage of cells with DNA bridges 72 h after induction of SAE knockdown was quantified using the microscopy approach described above, analyzing 100 cells per condition. The standard error of the mean was calculated from three independent experiments. A two-sided Student's *t*-test was performed. *$p$-values < 0.005

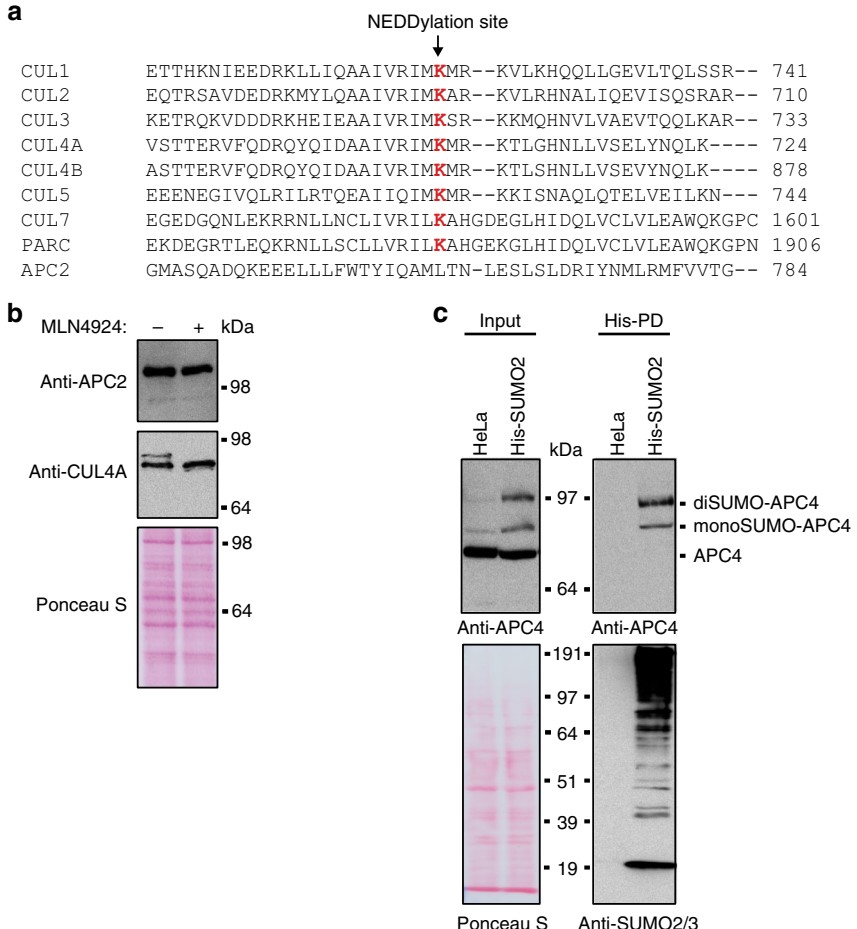

**Fig. 2** Posttranslational modification of the APC/C by SUMOylation. **a** Alignment of the Cullin domains present in the human Cullin proteins and the Cullin-like domain of APC2: The NEDDylated lysine residue within the Cullin proteins is highlighted in red and is absent in APC2. **b** HeLa cells were treated with DMSO as a negative control (−) or 1 μM MLN4924 (+), an inhibitor of the NEDD8-activating enzyme, for 16 h. Lysates were analyzed by immunoblotting with anti-APC2 and anti-CUL4A antibodies. Ponceau S staining was performed to monitor equal loading. **c** HeLa cells and HeLa cells expressing His-tagged SUMO2 (His-SUMO2) were subjected to a His-pulldown (His-PD). Input and pulldown samples were analyzed by immunoblotting making use of anti-APC4 and anti-SUMO2/3 antibodies. Blots were stained with Ponceau S to monitor equal loading. At least three independent experiments were performed and representative results are shown

transition from metaphase to anaphase. Taken together, both the results of the live cell imaging and the analysis of chromatin bridge formation clearly suggest that SUMOylation is particularly important for the proper segregation of chromatids during mitosis and for a regulated transition from metaphase to anaphase.

**The subunit APC4 of the APC/C is SUMOylated**. To increase our mechanistic understanding of the role of SUMO in mitotic progression, particularly during metaphase to anaphase transition, we aimed to identify relevant SUMO target proteins. SUMOylation is highly complex, co-regulating thousands of proteins[27]. An important regulator driving cell cycle progression from metaphase to anaphase is the APC/C, a Cullin-like ubiquitin ligase.

Alignment of the Cullin homology domain of all human Cullin subunits revealed that CUL1, CUL2, CUL3, Cul4A, Cul4B, CUL5, CUL7, and PARC harbor a NEDD8 acceptor lysine at a distinctive site required for activation[28] (Fig. 2a). However, the equivalent site is missing in APC2, the Cullin-like subunit of the APC/C, suggesting that this ubiquitin ligase is not regulated by NEDD8 in a canonical manner. This has been described before[29].

Immunoblot analysis of CUL4A in cell lysates showed a higher molecular weight band occurring in untreated cells and disappearing after blocking the NEDD8 E1 enzyme (Fig. 2b). Such a higher molecular weight band was not visible in the case of APC2, confirming that APC2 is indeed a Cullin-like subunit not being NEDDylated in vivo.

Our previous mass spectrometric analysis has identified APC4, a subunit of the APC/C, to be modified by SUMO[30]. However, the functional relevance of APC4 SUMOylation remained unknown. We now confirmed APC4 SUMOylation by performing pulldown experiments with HeLa cells expressing His-tagged SUMO2 (Fig. 2c). Immunoblot analysis of the pulldown fractions clearly showed SUMOylation of APC4 at two distinctive sites. We further observed similar SUMOylation patterns of APC4 by pulldown experiments in two additional human cell lines, U2OS and HT-1080 (Supplementary Fig. 2).

**Mutation of K772 and K798 in APC4 abolishes SUMOylation**. APC4 is a scaffolding subunit of the APC/C situated in the platform of the complex and has been shown to interact with several other APC/C subunits, such as APC1, APC2, APC5, APC8, and APC15[31]. By mass spectrometric analyses, we have

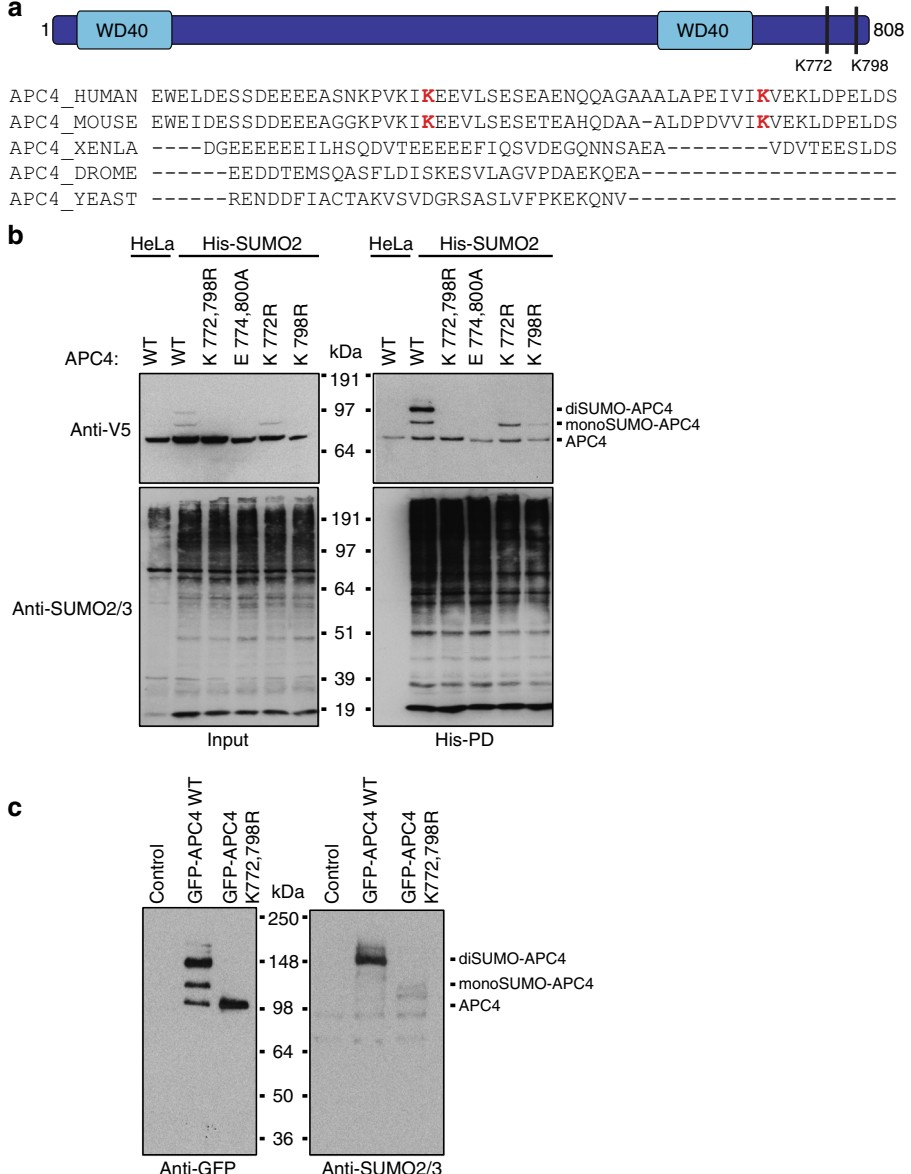

**Fig. 3** K772 and K798 of APC4 are the main SUMOylation sites within the APC/C. **a** Schematic overview of the APC4 subunit of the APC/C: The protein consists of 808 amino acids and contains a WD40-like domain split by a helix bundle domain. The lysine residues K772 and K798 identified to be SUMOylated (red) are situated at the C-terminal part of the protein. The C-terminal domain of human APC4 was aligned with the C-terminal domains of the homologs from *Mus musculus* (MOUSE), *Xenopus laevis* (XENLA), *Drosophila melanogaster* (DROME), and *Saccharomyces cerevisiae* (YEAST), demonstrating that the two SUMOylated residues are only conserved in human and mouse. **b** HeLa cells and HeLa cells expressing His-tagged SUMO2 (His-SUMO2) were transfected with V5-tagged versions of either APC4 wild type or different APC4 mutants. His-SUMO2 conjugates were purified by His-pulldown (His-PD). Input and pulldown samples were analyzed by immunoblotting with anti-V5 antibody. Equal SUMO2/3 levels were confirmed by immunoblotting with anti-SUMO2/3 antibody. Results shown are representative for two independent experiments. **c** HeLa cells were infected with control virus or retrovirus expressing either APC4 wild type or APC4 K772,798R mutant, both containing an N-terminal GFP-tag. The APC/C was purified from these cells by GFP-trap and SUMOylated in vitro. These samples were analyzed by immunoblotting with anti-GFP antibody to specifically visualize SUMOylation of APC4. To monitor the SUMOylation of the entire APC/C, samples were additionally analyzed with anti-SUMO2/3 antibody. Two independent experiments were performed and representative results are shown

previously found two specific lysine residues of APC4, K772, and K798 to be SUMOylated. These residues are both located at the very C-terminal part of the protein, which is only conserved in higher vertebrates (Fig. 3a). To study the relevance of APC4 SUMOylation for the function of the APC/C, we have generated two single (K772R and K798R) and two double APC4 mutants (K772,798R and E774,800 A) and transfected HeLa cells expressing His-tagged SUMO2 with wild-type (WT) and mutant constructs harboring an N-terminal V5 tag (Fig. 3b). Immunoblot

analysis of the His-pulldown samples confirmed that mutation of both K772 and K798 completely abolished SUMOylation of APC4, whereas only the di-SUMO-APC4 band was lost in the single mutants. As both SUMOylated lysine residues are situated in SUMO consensus motives, mutating E774 and E800 to alanine also prevented APC4 from being SUMOylated.

For further functional studies, we wanted to determine whether mutating K772 and K798 in APC4 indeed abolished APC/C SUMOylation or whether an additional subunit of the complex

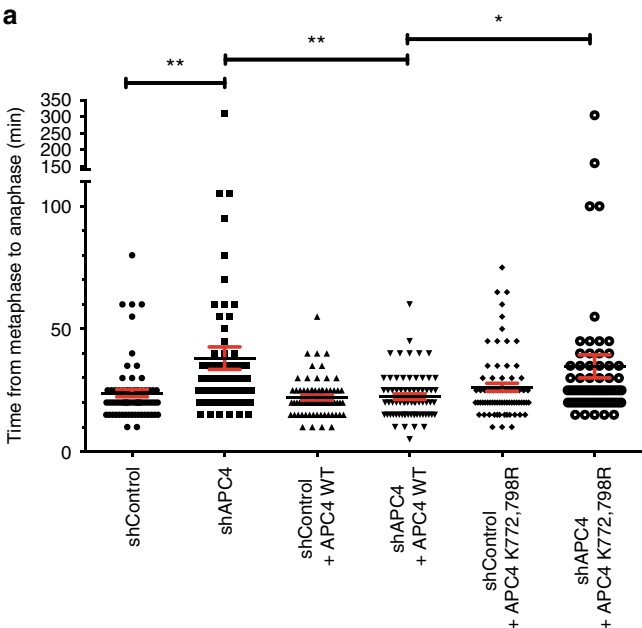

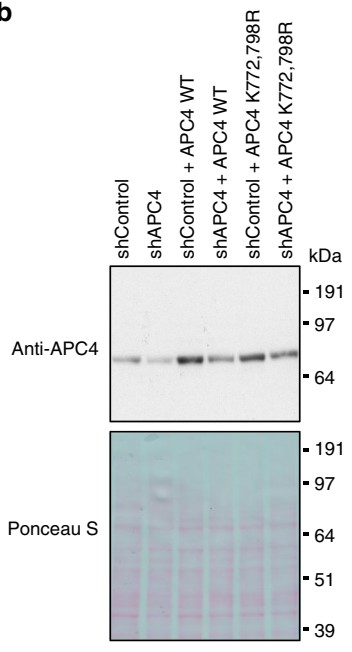

**Fig. 4** Mutation of K772 and K798 abolishes the rescue of a mitotic delay after knockdown of APC4. **a** HeLa cells stably expressing GFP-tagged Histone 2B and containing inducible constructs of either APC4 wild type or K772,798R mutant were infected with lentivirus expressing either a control shRNA or an shRNA raised against endogenous APC4. Knockdown of the inducible APC4 constructs was abolished by introducing point mutations at the sites of shRNA binding. Two days after infection with the viruses and induction of the APC4 constructs, the time needed for mitotic progression from metaphase to anaphase onset was monitored by live cell imaging. The mean value and the SEMs are shown for 70 cells per condition and $p$-values were obtained via Student's $t$-test. $p$-values: $*p < 0.02$, $**p < 0.005$. **b** Knockdown of APC4 was confirmed via western blot analysis of cell lysates 2 days after infection with the lentiviruses expressing either control shRNA or shRNA against APC4. The blot was stained with Ponceau S to monitor equal sample loading

could still be SUMOylated. Therefore, we expressed both APC4-WT and the APC4-K772,798R mutant fused to an N-terminal green fluorescent protein (GFP)-tag in HeLa cells. Immunoblot analysis revealed that both the WT and the mutant construct were equally expressed to nearly endogenous levels (Supplementary Fig. 3A). We purified the APC/C from these cells by a GFP-trap and SUMOylated these complexes in vitro. Immunoblot analysis demonstrated highly efficient SUMOylation of WT APC4, whereas the APC4-K772,798R double mutant could no longer be SUMOylated in vitro (Fig. 3c). Although we could confirm by silverstain, immunoblot analysis, and mass spectrometry (Supplementary Fig. 3B and C and Supplementary Data 1) that the other subunits of the APC/C were co-purified, only one strong band corresponding to SUMOylated APC4 is visible on a western blot analyzed with anti-SUMO2/3 antibody, suggesting that this subunit is indeed the dominant SUMO substrate in the complex. We specifically analyzed the potential SUMOylation of APC7, as this subunit has been identified to be SUMOylated in HeLa cells after heat shock treatment[32]. No SUMOylation was visible in vitro or in cells under control cell culture conditions (Supplementary Fig. 3C and Supplementary Fig. 4). However, we were able to confirm SUMOylation of APC7 after heat shock (Supplementary Fig. 4). These findings suggest that mutating K772 and K798 of APC4 indeed abolished SUMOylation of the entire APC/C.

**K772 of APC4 is located in a phospho-dependent SUMO motif.** Previous studies have described extensive crosstalk between phosphorylation and SUMOylation[34,35]. Phosphorylation events downstream of a SUMO site can strongly enhance SUMOylation and several of such phosphorylation-dependent SUMOylation motives (PDSMs) have been identified in previous site-specific mass spectrometric approaches on SUMO targets[30,36]. The tryptic peptide found for K772 in APC4 contained two phosphorylated serine residues further downstream, S777 and S779, proving that phosphorylation and SUMOylation are co-occurring events and suggesting that K772 is located in a PDSM (Supplementary Fig. 5A).

To test that SUMOylation of APC4 is dependent on preceding phosphorylation events, we mutated S777 and S779 into alanines, preventing them from being phosphorylated or into aspartates to mimic phosphorylation (Supplementary Fig. 5B). In addition, we mutated K798 to arginine to specifically investigate the effect on SUMOylation of K772. We transfected HeLa cells expressing His-tagged SUMO2 with the K798R,S777,779A and K798R, S777,779D triple mutants and used the K798R single mutant as a control. All constructs were fused to an N-terminal V5-tag to exclude endogenous APC4 from the analysis. Pulldown experiments revealed that, while the phosphomimic mutant did not alter SUMOylation, inhibiting phosphorylation of both S777 and S779 by mutating them into alanine resulted in decreased levels of K772 SUMOylation in vivo, indicating that K772 is indeed located in a PDSM.

**SUMOylation of APC4 is needed for mitotic progression.** We investigated whether SUMOylation of APC4 influences the function of the APC/C during mitotic progression. Knockdown of the APC4 subunit via small interfering RNA (siRNA) was shown to lead to a mitotic delay[33]. Therefore, we interfered with the complex through knockdown of its subunit APC4 via shRNA and were able to detect a significant delay in the transition from metaphase to anaphase as expected (Fig. 4a). While HeLa cells treated with control virus took on average 24 min to transit from metaphase to anaphase, cells expressing an shRNA against APC4 needed on average 38 min. Immunoblot analysis of cell lysates

after infection with lentivirus confirmed a decrease in APC4 expression after knockdown in contrast to cells treated with control shRNA (Fig. 4b). The delay in mitosis was rescued via expression of an exogenous APC4 WT construct that could not be targeted by the shRNA due to silent mutations. Cells expressing this construct needed on average 22 min and therefore showed a similar passage from metaphase to anaphase as the control cells. Rescue of the mitotic delay, however, was abolished, when expressing an exogenous APC4 K772,798R double mutant construct. These cells needed approximately 35 min to proceed with mitosis, suggesting that SUMOylation of these two residues is needed for a correct passage from metaphase to anaphase (Fig. 4a). Immunoblot analysis showed that both exogenous constructs were equally expressed at levels close to endogenous APC4 (Fig. 4b).

**APC4 SUMOylation increases ligase activity toward Hsl1**. Next we asked whether SUMOylation of APC4 affected the ubiquitin ligase activity of the APC/C. Therefore, we set up an in vitro system to test endogenous APC/C purified from HeLa cells. First, the purified APC/C was SUMOylated in vitro. As a negative control, we treated endogenous APC/C with the SUMOylation machinery lacking the SUMO conjugating enzyme UBC9. After subsequent washes to remove the SUMO machinery, the recombinant coactivator CDH1 and a fragment of the APC/C model substrate Hsl1[37] were bound to the complex. Finally, an in vitro ubiquitylation assay was performed both with the control and the SUMOylated APC/C (Fig. 5a). Silverstain and mass spectrometric analysis of the purified APC/C confirmed the presence of the core subunits (Fig. 5b, Supplementary Data 2). Furthermore, the unmodified form of APC4 was hardly visible on the silverstain if the purified complex was incubated with the SUMOylation machinery including the SUMO conjugating enzyme UBC9, suggesting that SUMOylation of APC4 on the beads was highly efficient. This was further confirmed by immunoblot analysis (Fig. 5c).

Next, we performed in vitro ubiquitylation assays to determine the activity of the complex without and with SUMOylation of APC4 and monitored ubiquitylation of Hsl1 over time by immunoblot (Fig. 5d). Immunoblot analysis of APC2, the Cullin-like subunit of the complex, and APC11, the RING-box containing subunit, confirmed equal levels of the APC/C during the reaction. While ubiquitylation of Hsl1 by the non-SUMOylated complex was only moderate after 30 min, the SUMOylated APC/C showed a much higher efficiency in ubiquitylating Hsl1. A similar effect was observed when SUMOylating the complex with a SUMO mutant, where all lysines are mutated to arginines, suggesting that SUMO chain formation is not involved in this process (Supplementary Fig. 6). Therefore, we conclude that SUMOylation of APC4 enhances the ubiquitin ligase activity of the complex in vitro toward the model substrate Hsl1.

**APC4 SUMOylation enhances binding of KIF18B to the complex**. We were interested whether SUMOylation would have a similar effect on endogenous APC/C substrates from human cells. Therefore, we performed a mass spectrometric analysis to identify proteins preferentially binding to SUMOylated APC/C. To achieve this, we SUMOylated recombinant APC/C in vitro, re-purified the APC/C via its strep-tagged APC4 subunit, added HeLa lysate to co-immunoprecipitate binding partners, and identified these proteins by label-free quantification (LFQ) analysis. As a negative control, we performed the same experiment with recombinant APC/C containing the APC4 K772,798R double mutant (Fig. 6a). Immunoblot analysis revealed that the

entire amount of WT APC4 was modified by two SUMO moieties, while the mutant APC4 was not SUMOylated (Fig. 6b). Mass spectrometric analysis confirmed the presence of all core subunits of the APC/C after addition of HeLa lysate and revealed no significant difference between the subunits of the complex containing WT APC4 versus the mutant form (Fig. 6c, Supplementary Data 3), indicating that SUMOylation does not change the composition of the complex. Only for ANAPC11, the LFQ ratio between WT and mutant complex was elevated. However, according to the p-value this is not a significant result and only two peptides were identified in our mass spectrometric analysis, since ANAPC11 is a very small protein of 10 kDa. In addition to the subunits of the complex, we found at least 13 known APC/C substrates in our mass spectrometric screen, including the kinesins KIF2C and KIF4A, but these kinesins did not show any significant change in binding to the SUMOylated APC/C. We identified another kinesin, KIF18B, to be significantly enriched in the samples containing the SUMOylated WT complex, suggesting that this protein might be preferentially regulated by SUMOylated APC/C (Fig. 6d, Supplementary Data 3).

**Ubiquitylation of KIF18B is enhanced after APC4 SUMOylation**. As expected, immunoblot analysis revealed that KIF18B co-immunoprecipitated more efficiently with SUMOylated APC/C than with the non-SUMOylated mutant complex (Fig. 7a). Equal amounts of the APC/C in both samples was confirmed via mass spectrometry (Supplementary Data 3) and immunoblotting against the subunits APC4 (Fig. 6b) and APC7 (Supplementary Fig. 7A). We further analyzed whether KIF18B is able to bind to SUMO itself. Therefore, we incubated HeLa cells with a His-tagged SUMO2 construct mimicking a SUMO chain consisting of three SUMO2 moieties (3xSUMO2). Immunoblot analysis confirmed that KIF18B strongly bound to these SUMO chains (Fig. 7b). Treatment of HeLa cells with an siRNA against KIF18B verified that the antibody used in these experiments indeed recognizes endogenous KIF18B (Supplementary Fig. 7B).

We further wanted to identify specific KIF18B peptides responsible for SUMO binding and tested 12 putative SUMO interaction motive (SIM)-containing peptides and their respective SIM mutant peptides for binding to recombinant 3xSUMO2 chains in an anti-SUMO2/3 ELISA assay (Supplementary Fig. 7C). To confirm the applicability of this assay, we used two well-characterized SIM-containing peptides of the SUMO chain binding protein RNF4 and the respective SIM mutant peptides as controls. Indeed, we could observe strong binding of the SIM-containing peptides of RNF4 to the SUMO chains, while mutation of the SIMs abolished binding (Fig. 7c). Furthermore, 11 of the KIF18B peptides we tested did not bind to SUMO chains (Supplementary Fig. 7C). However, we could detect binding of the KIF18B peptide containing the putative SIM LLALI (aa 276–290) to SUMO. This interaction was significantly reduced for its respective SIM mutant, where all hydrophobic residues were mutated to alanines (Fig. 7c), suggesting that this motif is involved in the binding of KIF18B to SUMO.

Next, we performed in vitro ubiquitylation experiments with recombinant WT or mutant APC/C and Flag-tagged KIF18B (Fig. 7d). These experiments identified KIF18B as a novel ubiquitylation target of APC/C. However, the non-SUMOylated WT complex and the mutant APC/C treated with the SUMOylation machinery both showed minor ubiquitylation activity, resulting in only one higher molecular weight band visible on unmodified KIF18B. In contrast to this, ubiquitylation by the SUMOylated WT complex was strongly increased leading to at least two higher molecular weight bands visible above the unmodified form of KIF18B. These bands did not disappear when

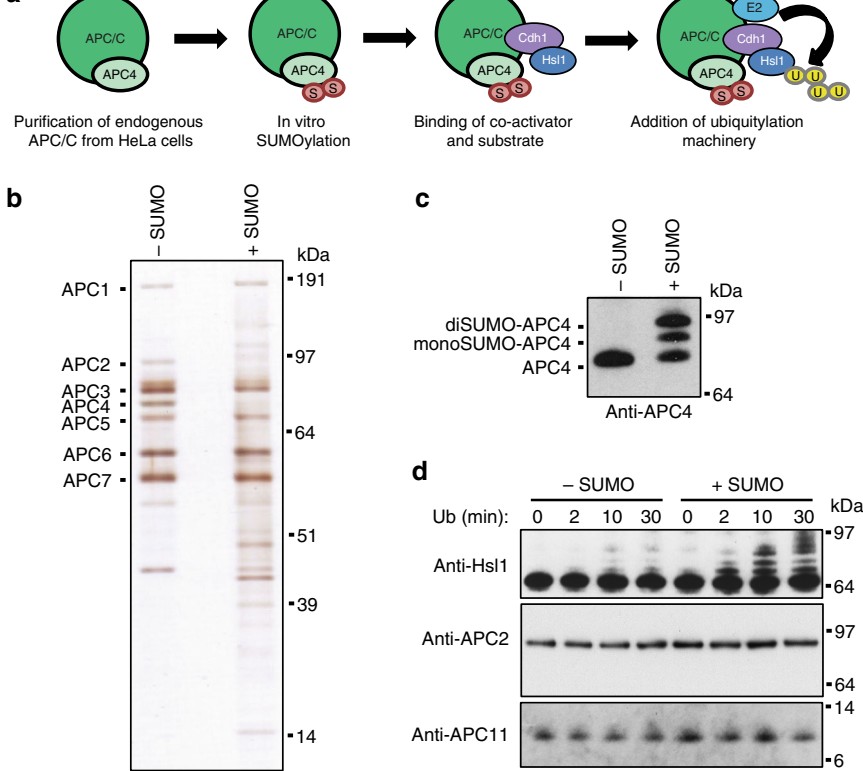

**Fig. 5** SUMOylation of the endogenous APC/C enhances its ubiquitylation efficiency toward the model substrate Hsl1 in vitro. **a** Schematic overview on the experimental approach: Endogenous APC/C was purified from HeLa cells making use of beads coupled with anti-CDC27 antibody. The purified complex was SUMOylated in vitro. As a negative control, half of the purified complex was incubated with the SUMOylation machinery leaving out the SUMO-conjugating enzyme UBC9. Afterwards, recombinant Cdh1, a co-activator of the APC/C, and recombinant Hsl1, an APC/C substrate, were added to the samples. Finally, in vitro ubiquitylation assays were performed. **b** The APC/C was purified from HeLa cells by immunoprecipitation and SUMOylated in vitro. Both the negative control without UBC9 (−SUMO) and the SUMOylated APC/C (+SUMO) were silverstained to visualize the presence of the different APC/C subunits. **c** Endogenous APC/C was purified from HeLa cells as described before and either SUMOylated or mock-treated without UBC9. SUMOylation of APC4 was confirmed by immunoblotting with anti-APC4 antibody. **d** The ubiquitylation activity of the non-SUMOylated control and the SUMOylated APC/C purified as described before were tested in vitro. The SUMOylation machinery was removed by washing the beads and recombinant Cdh1, a coactivator of the APC/C, and Hsl1, a model substrate from *Saccharomyces cerevisiae*, were bound to the complex. Then the complex was eluted from the beads by peptide elution and the ubiquitylation machinery was added to the reaction. Samples were taken at different timepoints after starting the ubiquitylation reaction and were analyzed by immunoblotting with anti-Hsl1 antibody to visualize ubiquitylation levels of Hsl1. To monitor equal levels of the APC/C, the same samples were analyzed with anti-APC2 and anti-APC11 antibodies. This figure shows results representative of at least three independent experiments

treating the samples with the SUMO protease SENP2 as the final step of the assay, confirming that these bands indeed result from the ubiquitylation reaction and are not SUMOylated forms of KIF18B. As a control, SUMOylation of APC4 was monitored by immunoblotting (Supplementary Fig. 7D). We further tested the known APC/C substrate Securin in similar in vitro ubiquitylation assays. However, no difference between the ubiquitylation activity of non-SUMOylated WT complex, SUMOylated WT complex, or mutant complex was visible in the case of Securin (Supplementary Fig. 7E), demonstrating that KIF18B but not Securin is preferentially recognized by SUMOylated APC/C, thereby driving ubiquitylation of the substrate (Fig. 7e).

## Discussion
We found that the posttranslational modifier SUMO is essential for proper mitotic progression and showed that, similar to mice[24], depletion of the SUMO conjugation machinery delayed progression from metaphase to anaphase in human cancer cells and led to severe chromosomal segregation defects (Fig. 1). Only little is known about the target proteins regulated by SUMO during

chromosomal segregation. Relevant SUMO target proteins include Topoisomerase II and Polo-like kinase 1-interacting checkpoint helicase[38–41]. Here we identified the ubiquitin E3 ligase APC/C as a key SUMO target. Since the APC/C has been previously found as a main guarantor for anaphase onset and accurate chromosomal segregation, it represents a major SUMO target to explore in the context of the mitotic defects observed after knockdown of the SUMOylation machinery[8,30,42].

SUMOylation of APC4 increases during mitosis, but so far the effect of SUMOylation on the function of human APC/C has been unresolved[42]. Here we identified two specific residues at the very C-terminus of APC4, K772 and K798, as SUMO acceptor sites (Fig. 3) and showed that SUMOylation of these two residues is needed for an efficient cellular progression from metaphase to anaphase (Fig. 4). These findings might therefore partly explain the mitotic phenotypes observed after knockdown of the SAE (Fig. 1).

We further were able to detect an increased ubiquitylation activity of the SUMOylated APC/C in vitro (Figs. 5d and 7d). This effect appeared to be restricted to a subset of APC/C target proteins. While ubiquitylation of the yeast substrate Hsl1 and the

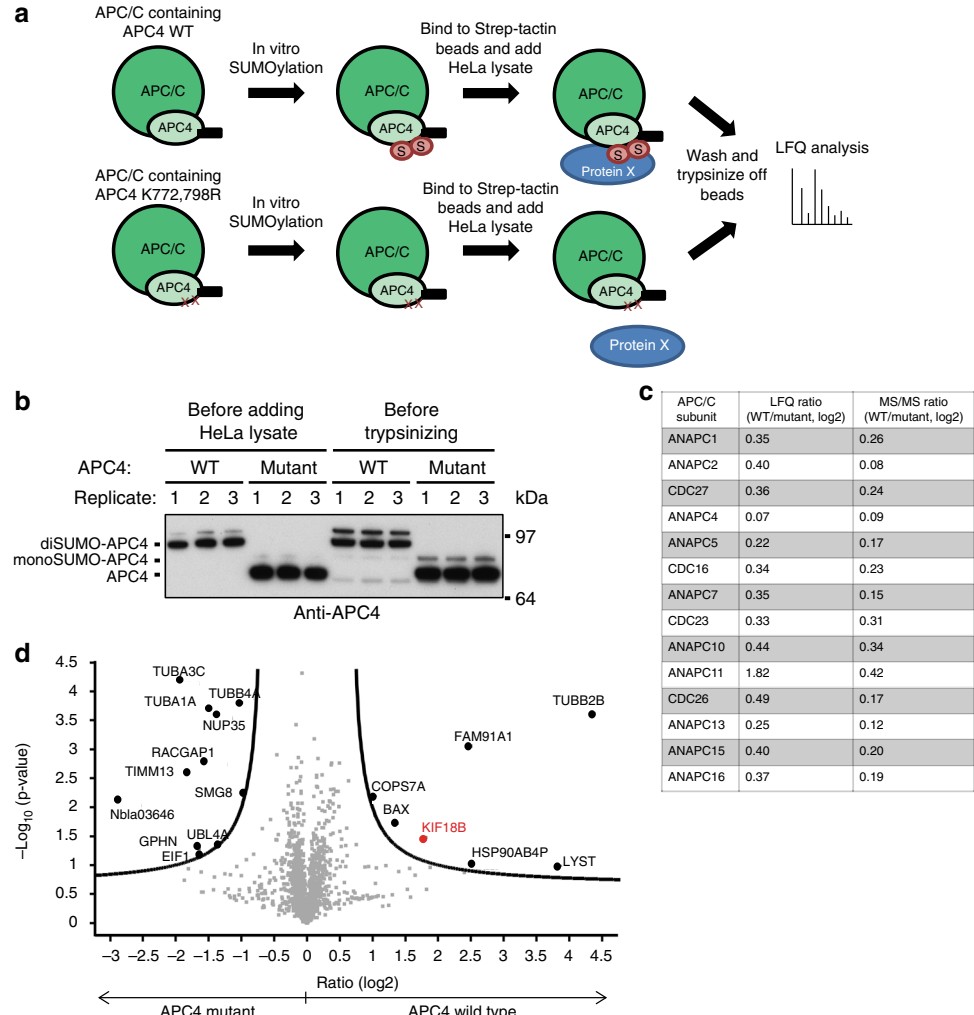

**Fig. 6** Mass spectrometric analysis identifies KIF18B to preferentially bind to SUMOylated APC/C. **a** Schematic overview on the mass spectrometric approach designed to identify differences in the interactomes of SUMOylated versus non-SUMOylated APC/C: Recombinant APC/C containing either APC4 wild type or APC4 K772,798R mutant, was SUMOylated in vitro. The complex was bound to Strep-tactin beads via a C-terminal Strep-tag on APC4 and the SUMOylation machinery was washed away. HeLa cells were lysed in buffer containing NEM to inhibit SUMO proteases and lysates were added to the wild-type and mutant complexes. After mild washing, proteins were trypsinized and the resulting peptides were subjected to mass spectrometry. Data were obtained from three biological repeats measured in three technical repeats each. **b** To monitor the SUMOylation levels of APC4 wild type (WT) and APC4 K772,798R mutant before and after adding the HeLa lysate, samples were taken at the respective steps and immunoblotted with anti-APC4 antibody. **c** The table depicts the different subunits of the APC/C identified in the mass spectrometric analysis and the respective label-free quantification (LFQ) and MS/MS (spectral counting) ratios in log2, visualizing that there is no significant difference between the composition of the wild-type complex (WT) and the complex containing the APC4 SUMOylation-deficient mutant. **d** Volcano plot showing the log 2 ratio of the protein intensities obtained by purification with wild-type APC4 compared to the intensities obtained with mutant APC4 on the x axis versus the negative p-value ($\log_{10}$) calculated from a permutation-based false discovery rate on the y axis. Proteins that were either upregulated in the wild-type samples or upregulated in the mutant samples are displayed as large black dots. KIF18B, shown in red, is upregulated in the wild-type sample, indicating that it preferentially bound to SUMOylated APC/C

novel APC/C target KIF18B was increased, ubiquitylation of Securin was not altered (Supplementary Fig. 7E).

Consistent with our findings, degradation of several APC/C target proteins in yeast has been described to decrease after depletion of the SUMO-conjugating enzyme Ubc9 and the SUMO homolog Smt3, indicating that regulation of the APC/C by SUMOylation might be conserved throughout the eukaryotic kingdom[43]. However, this effect was also observed for the yeast homolog of Securin, Pds1, suggesting that the impact of SUMOylation on APC/C function in yeast might differ from the impact of SUMOylation in human cells. This could be explained by the fact that none of the APC/C subunits in yeast have been identified to be SUMOylated. Consistently, the SUMOylated lysine residues in human APC4 are not conserved in yeast APC4.

We therefore hypothesize that SUMOylation might affect the APC/C activity in yeast in an alternative or indirect manner.

The observation that SUMOylation of the human APC/C only affects Hsl1 and KIF18B in our experiments further raises the question how SUMOylation specifically regulates the recruitment of a distinct set of APC/C targets. Since both Hsl1 and KIF18B contain putative SIMs and such a motif is not predicted for Securin, one explanation for the increased ubiquitylation of specific APC/C targets might be the enhanced recruitment to the complex by SUMO–SIM interactions[44]. A multitude of known APC/C substrates are predicted to contain SIMs and might therefore be differentially regulated after SUMOylation of the APC/C via SUMO–SIM interactions, while other substrates might not be affected (Supplementary Table 1). We have identified 12

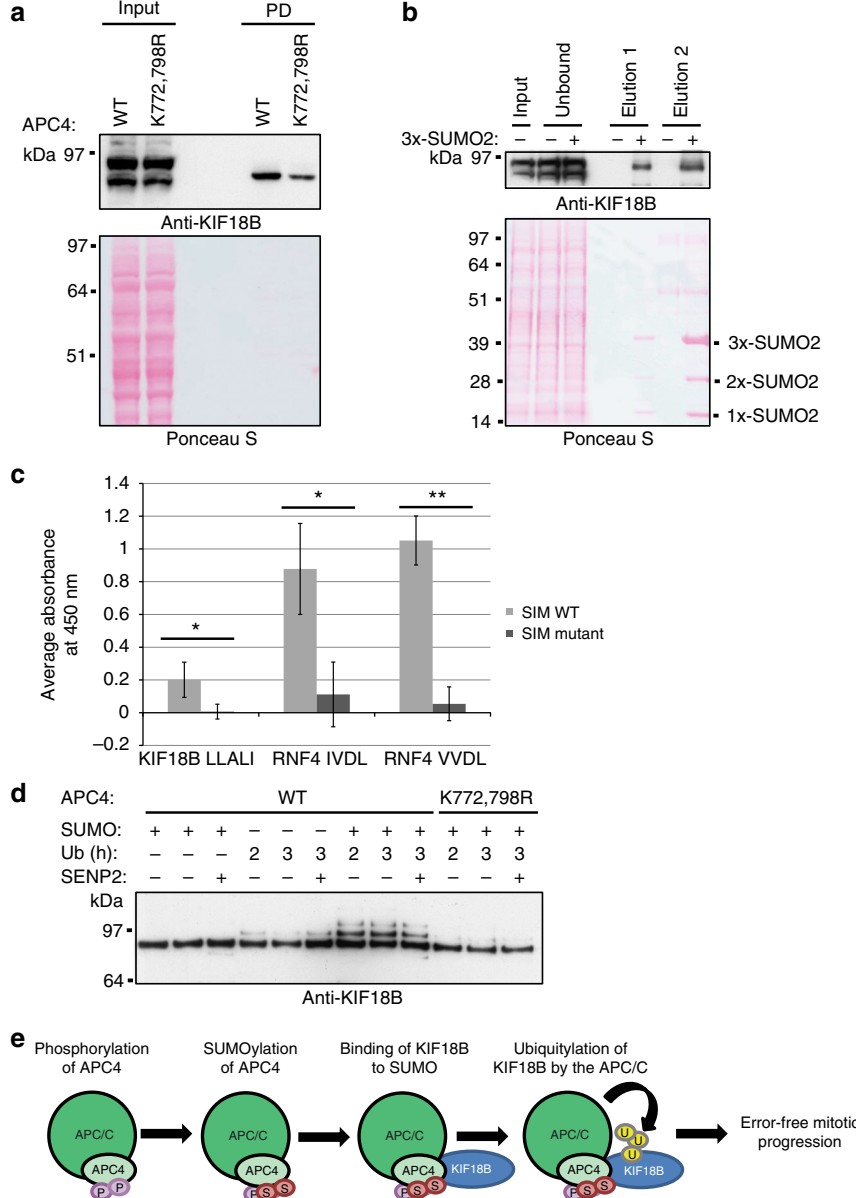

**Fig. 7** Binding of KIF18B to SUMOylated APC/C enhances its ubiquitylation. **a** HeLa lysate and samples taken after co-purification of potential binders with the Strep-tagged APC/C wild type and mutant were analyzed by immunoblotting with anti-KIF18B antibody. **b** Recombinantly expressed His-tagged tri-SUMO2 chains (3xSUMO2) were immobilized on beads and incubated with a HeLa lysate. After washing, bound proteins were first eluted with 8 M urea (elution 1) and in a second step with 8 M urea including imidazole (elution 2). To determine whether KIF18B co-purified with 3xSUMO2, input, unbound, and pulldown samples were analyzed by immunoblotting with anti-KIF18B antibody. Data shown are representative of two individual experiments. **c** The KIF18B peptide NINRSLLALINVLNA and its  corresponding putative SIM mutant NINRSAAAAANVLNA fused to biotin and a PEG linker were immobilized on avidin-coated wells and incubated with 3xSUMO2 chains. Afterwards, an anti-SUMO2/3 ELISA was performed. As a control, SIM-containing peptides of RNF4 and their respective SIM mutant peptides were tested. Mean values and standard deviations were calculated from three technical replicates and a two-sided Student's $t$-test was performed. p-values: *$p < 0.05$, **$p < 0.005$. Representative results of two independent experiments are shown. **d** Recombinant APC/C either containing APC4 wild type or APC4 K772,798R mutant was SUMOylated in vitro or treated similarly without UBC9 as a negative control. The co-activator CDH1 and Flag-tagged KIF18B were bound to the complex in vitro and the ubiquitylation machinery was added. The ubiquitylation reaction was either stopped after 2 or after 3 h as indicated. Additionally, SENP2 was added to a third sample after 2 h and the reaction was run for another hour to analyze whether higher molecular weight bands are indeed the result of ubiquitylation and not SUMOylation of KIF18B. KIF18B was visualized by immunoblotting with the anti-KIF18B antibody. This experiment was repeated three times and representative results are shown. **e** Schematic overview of the effect of APC4 modification on APC/C function: SUMOylation enhances the recruitment of KIF18B to the APC/C. Thereby, ubiquitylation of KIF18B is facilitated, contributing to error-free progression through mitosis

putative SIMs in KIF18B, and indeed the protein is binding more efficiently to SUMOylated APC/C than to the non-SUMOylated control in our mass spectrometric analysis (Figs. 6d and 7a). Furthermore, KIF18B binds to a construct mimicking SUMO2 chains in our co-immunoprecipitation (Co-IP) experiments with

HeLa cells (Fig. 7b). Additionally, we could show that a short KIF18B peptide containing one of the putative SIMs bound to SUMO chains in a SIM-dependent manner, suggesting that this motif is involved in binding to SUMOylated APC/C (Fig. 7c).

KIF18B has been described to promote microtubule depolymerization and might thereby regulate chromosomal alignment and segregation[45]. Deregulation of KIF18B could therefore contribute to mitotic defects observed for downregulation of the SUMO machinery. Furthermore, other APC/C substrates might be deregulated after knockdown of the SUMO machinery or mutagenesis of K772 and K798 in APC4. In particular, other kinesins have been identified as APC/C substrates[46,47] and contain putative SIMs (Supplementary Table 1). Therefore, the effect on ubiquitylation of KIF18B is unlikely to be the only explanation for the severe mitotic phenotypes observed.

Interestingly, we identified another known APC/C substrate, RacGAP1[48], to show decreased binding to SUMOylated APC/C in our mass spectrometric screen (Fig. 6d). Therefore, the effect of SUMOylation of the APC/C on each individual substrate needs to be carefully addressed since it can lead to enhanced or reduced substrate binding.

The APC/C regulates its substrates in a timely order rather than simultaneously and this order is essential for accurate mitotic progression[49,50]. Whether SUMO contributes to this orderly fashion of APC/C activity remains to be investigated.

Recently, PTM of APC1 by phosphorylation has been shown to result in a conformational change of an autoinhibitory domain facilitating the binding of the co-activator Cdc20[51]. Elaborate structural analysis of the human APC/C has revealed that APC4 is in contact with several subunits, including the Cullin domain of APC2[31,52]. However, the C-terminus of APC4, including the SUMOylated lysine residues identified here, is disordered and therefore missing in the structures obtained by crystallography as well as in EM density maps. This suggests that this domain is highly flexible and might therefore indeed be able to induce conformational changes influencing the interaction with other APC/C subunits or additional binding partners after SUMOylation. It would be interesting to investigate whether SUMOylation of APC4 would enable the structural resolution of the C-terminal domain through enhanced interaction with other subunits of the complex. Several APC/C subunits shown to interact with APC4, such as APC1, APC2, APC5 and APC8, contain predicted SIMs that could facilitate a structural change upon modification[44].

Additionally, we could show that K772 of APC4 is located in a phosphorylation-dependent SUMOylation motif (Supplementary Fig. 5). This dependency on phosphorylation was found for other SUMO target proteins, demonstrating crosstalk between phosphorylation and SUMOylation[30,36]. Phosphorylation of S777 and S779 was found in both in vitro and in vivo studies[12,53–55], and indeed, mass spectrometric analysis of SUMOylation sites in HeLa cells has identified phosphorylation of S777 and S779 occurring simultaneously with SUMOylation of K772 on APC4[30]. These results suggest that phosphorylation of APC4 might be a prerequisite to SUMOylation and needed for the regulation of the ubiquitin ligase activity.

In conclusion, we show that inhibition of the SUMO-activating and the SUMO-conjugating enzyme caused a mitotic delay and severe chromosomal defects, which could enhance tumor development. Several components of the SUMOylation machinery are deregulated in various types of cancer and are thus of great interest in cancer research[56]. Efforts are ongoing to generate inhibitors of the SUMO machinery to block proliferation of cancer cells[57–62]. The proper application of these inhibitors in the clinic requires a deep understanding of the functional roles of SUMO in cell cycle progression. The identification of the mitotic group of SUMO targets will be helpful to understand how SUMO regulates mitotic progression by coordinating these targets in a group-like manner[63].

In this study, we describe a novel mechanism by which SUMOylation regulates one of the major mitotic driving forces, the ubiquitin ligase APC/C. This enhances our understanding of mitotic cooperativity of PTMs and could be helpful for future cancer drug development.

## Methods

**Plasmids**. To create the inducible SAE knockdown constructs, forward (FW) and reverse (RV) oligonucleotide sequences were designed to contain either a scrambled shRNA or a target shRNA sequence flanked by *Bbs*I and *Xho*I restriction sites: ishControl_FW:TCCCGAGGATAGACGCTTTAAATAATTCAAGAGATTA TTTAAAGCGTCTATCCTCTTTTTC,ishControl_RV:TCGAGAAAAAGAGG ATAGACGCTTTAAATAATCTCTTGAATTATTTAAAGCGTCTATCCTC, ishSAE1_FW:TCCCGCTATGTTGGTCCTTTGTTTATTCAAGAGATAAACAA AGGACCAACATAGCTTTTTC,ishSAE1_RV:CTCGAGAAAAAGCTATGTT GGTCCTTTGTTTATCTCTTGAATAAACAAAGGACCAACATAGC, ishSAE2.1_FW:TCCCGCTGTATTGAAAGTAGGAATATTCAAGAGATATTCC TACTTTCAATACAGCTTTTTC,ishSAE2.1_RV:CTCGAGAAAAAGCTGTATT GAAAGTAGGAATATCTCTTGAATATTCCTACTTTCAATACAGC,ish-SAE2.2_FW:TCCCGCACCAGATGTCCAAATTGAATTCAAGAGATTCAA TTTGGACATCTGGTGCTTTTTC,ishSAE2.2_RV:CTCGAGAAAAAGCACCAG ATGTCCAAATTGAATCTCTTGAATTCAATTTGGACATCTGGTGC). FW and RV oligonucleotides were annealed and ligated into the pH1tet-flex backbone[64] digested by *Bbs*I and *Xho*I restriction enzymes (Biolabs). Plasmids for the scrambled shRNA were digested with *Pac*I restriction enzyme (Biolabs) and ligated into the *Pac*I digested and dephosphorylated FH1t_UTG backbone[64]. Plasmids for the target shRNAs were digested with *Nhe*I and *Bst*BI restriction enzymes (Biolabs) and ligated into the *Nhe*I and *Bst*BI digested and dephosphorylated FH1t_UTG_Neo backbone. Constitutive knockdown of SAE2 and UBC9 in HCT116 and HT-1080 cells was achieved by the expression of the following shRNAs: shSAE2.1:CCGGGCTGTATTGAAAGTAGGAATACTCGAGTATTC CTACTTTCAATACAGCTTTTT,shSAE2.2:CCGGGCACCAGATGTCCAAATT GAACTCGAGTCAATTTGGACATCTGGTGCTTTTT,shUBC9.1:CCGGGC CTACACGATTTACTGCCAACTCGAGTTGGCAGTAAATCGTGTAGGCTTT TT,shUBC9.2:CCGGTGGAGGAAAGACCACCCATTTCTCGAGAAATGGG TGGTCTTTCCTCCATTTTT.

Knockdown of endogenous APC4 was performed using the following shRNA: CCGGGCTGGTACTTGTCTTGCATTACTCGAGTAATGCAAGACAAGT ACCAGCTTTTT.

The cDNA of ANAPC4 was obtained from Invitrogen/Life Technologies (ultimate ORF clone IOH54154) in the donor vector pENTR221. For fusions with an N-terminal V5-tag, the cDNA was cloned into pcDNA3.1/nV5-DEST using standard GATEWAY technology (Invitrogen). N-terminal GFP-fusions were obtained by GATEWAY cloning into the retroviral vector pBabe-GFP-puro-DEST (a kind gift of M. Timmers and P. de Graaf, Utrecht, The Netherlands). Mutants of APC4 were produced via site-directed mutagenesis of the entry clone.

Inducible constructs of APC4 WT and APC4 K772,798R mutant were obtained by GATEWAY cloning into pCW57.1. Resistance to the shRNA against APC4 was achieved via site-directed mutagenesis of the APC4 entry clones.

The pHIS-TEV30a-3xSUMO2 construct was a kind gift of Dr. R.T. Hay[65]. The N-terminal His6-tag was changed into a His10-tag by site-directed mutagenesis. The pBIOBS-KIF18B-Flag construct was published previously[45]. Constructs to purify a His-tagged fragment of Hsl1 (aa 667-872) and His-tagged UbcH10[66] were a kind gift from Dr. J.M. Peters, Vienna, Austria.

**Cell lines and cell culture**. HT-1080, HCT116, and U2OS cells were obtained from ATCC and HeLa cells originated from the laboratory of René Medema at the Netherlands Cancer Institute. All cell lines have been authenticated via STR profiling using 10 different markers and have been tested to be free of mycoplasma.

HCT116 cells were cultured in 45% RPMI1640+Glutamax (Gibco, Invitrogen Corporation, Grand Island, NY, USA), 45% Dulbecco's modified Eagle's medium/F12 (Gibco) and 10% fetal calf serum (FCS; Gibco), complemented with 100 U ml$^{-1}$ penicillin and 100 mg ml$^{-1}$ streptomycin (Gibco). HeLa, U2OS, and HT-1080 cells were cultured in Dulbecco's modified Eagle's medium (Gibco) including 10% FCS (Gibco) and 100 U ml$^{-1}$ penicillin and 100 mg ml$^{-1}$ streptomycin (Gibco). HeLa cell lines for inducible knockdown were cultured in the same medium supplemented with 300 μg ml$^{-1}$ Geneticin (G418; Life technologies). To induce the expression of the shRNAs, these HeLa cell lines were treated with 10 ng ml$^{-1}$ doxycycline for the indicated amount of time. For inhibition of the NEDD8-activating enzyme, HeLa cells were treated with 1 μM MLN4924 for 16 h. Heat shock was done for 75 min at 43 °C. For recovery samples, the cells were heat shocked as above but then put back at 37 °C for 2 h before lysing.

**Antibodies**. The following primary antibodies were used at a dilution of 1:1000: rabbit anti-APC2 (Cell Signaling Technology, CST12301), rabbit anti-CUL4A

(Bethyl, A300-739A), rabbit anti-APC4 (Bethyl, A301-176A), rabbit anti-V5 (Abcam, ab9116), mouse anti-SUMO2/3 (Abcam, ab81371), mouse anti-SUMO2/3 (MBL-Sanbio, M114-3), rabbit anti-GFP (Sigma, 1814460), mouse anti-His (Sigma, H1029), rabbit anti-SART1 (custom-made by Eurogentec)[67], rabbit anti-CDC27 (Santa Cruz, sc9972), rabbit anti-CDH1, rabbit anti-Hsl1 and rabbit anti-APC11 (kind gifts from Dr. J.M. Peters, Vienna)[66], rabbit anti-KIF18B (Bethyl, A303-982A), and rabbit anti-Securin (Cell Signaling Technology, CST13445).

**Transfection of cells and virus infection**. Cells were transfected at 60% confluency in a 15-cm dish with 24 μg plasmid DNA and 60 μg polyethylenimine (Alfa Aesar). Cells were infected with lentivirus and retrovirus at a multiplicity of infection (MOI) of 3. To obtain stable cell lines, selection was started 24 h after infection using 600 μg ml$^{-1}$ Geneticin (G418; Life Technologies) for the inducible knockdown of SAE and 2.5 μM puromycin (Calbiochem) for cells infected with the retroviral GFP-APC4 constructs. To analyze the knockdown of KIF18B, HeLa cells were transfected with 20 nM of control siRNA (#D-001210-01-05, Dharmacon) or siRNA against KIF18B (#J-010460-11-0005, Dharmacon) by mixing the siRNAs with 200 μl Optimem (#31985070, Life Technologies) and 5 μl Dharmafect (#T-2001-02, GE Life Sciences), incubating for 20 min at room temperature, complementing with 1.6 ml Dulbecco's modified Eagle's medium (Gibco) including 10% FCS (Gibco), and adding the mix to the cells. Cells were lysed in SNTBS (2% sodium dodecyl sulfate (SDS), 1% N-P40, 50 mM Tris pH 7.5, 150 mM NaCl) for western blot analysis 72 h after transfection.

**Microscopy**. In case of the inducible SAE knockdown, stable HeLa cell lines were plated on day 0 in six-well plates. Knockdown of the target protein was induced by adding medium containing 10 ng ml$^{-1}$ doxycycline on day 1. Cells were split on day 3 onto coverslips in 24-well plate wells for microscopy and into 6-well plate wells for immunoblot analysis to confirm knockdown efficiency. On day 4, cells on coverslips were fixed with 3.7% formaldehyde in phosphate-buffered saline (PBS) for 15 min at room temperature and washed five times with PBS. Cells were dehydrated by subsequently washing the coverslips for 1 min with 70%, 90%, and 100% ethanol. Finally, coverslips were dried and placed on a slide using ProLong Gold (Life Technologies) containing 5 μg ml$^{-1}$ Hoechst 33258 (Life Technologies). Cells in 6-well plates were lysed in SNTBS (2% SDS, 1% N-P40, 50 mM Tris pH 7.5, 150 mM NaCl) for western blot analysis. To analyze the constitutive knockdown of SAE2 and UBC9 in HCT-116 and HT-1080 cells, cells were plated onto coverslips in 24-well plate wells for microscopy and into 6-well plate wells for immunoblot analysis to confirm knockdown efficiency 1 day after lentivirus infection. After 48 h, cells on coverslips were fixed and stained for microscopy as described above and samples for immunoblot analysis were lysed in SNTBS.

**Live cell imaging to analyze knockdown of SAE1**. Stable HeLa cell lines were plated on day 0 in six-well plates. Knockdown of the target protein was induced by replacing the medium with medium containing 10 ng ml$^{-1}$ doxycycline on day 1. After 40 h of induction, cells were split into 8-well glass bottom dishes (Labtek). Six hours later, SiR-DNA (Spirochrome) was added at a final concentration of 250 nM in Leibovitz L15 CO$_2$-independent medium (Gibco). After 2 h incubation, live cell imaging was started on a DeltaVision microscope (Deltavision Elite; Applied Precision). Images were acquired every 5 min using a 20× NA 0.25 air objective (Olympus) and a high-resolution CCD camera (Coolsnap HQ2; Photometrics). Z-stacks were acquired with 3.33 μm intervals. Images were processed using Softworx (Applied Precision) and Image J software. Projections of Z-stacks were made and analyzed by marking the frames of NEB metaphase, and anaphase for each cell going through mitosis. Subsequently, the amount of time needed from NEB until metaphase and from metaphase until anaphase was calculated for each dividing cell. Averages and standard deviations were calculated for >200 cells per condition resulting from three independent experiments. Statistical analysis was performed using a two-sided Student's t-test.

**Live cell imaging in rescue experiment**. HeLa cells expressing GFP-tagged Histone 2B[68] and harboring an inducible construct of either APC4 WT or APC4 K772,798R mutant were seeded in a 6-well plate (125,000 cells per well). On day 2, expression of the APC4 constructs was induced by adding medium containing 10 ng ml$^{-1}$ doxycycline and cells were infected with lentivirus expressing either a control shRNA or an shRNA raised against APC4 at an MOI of 3. Cells were split for microscopy into a 24-well plate on day 3 (40,000 cells per well) and subjected to live cell imaging on day 4. The amount of time needed for the dividing cell to proceed from metaphase to anaphase onset was monitored. SEMs were calculated for 70 cells per condition and statistical analysis was performed using a two-sided Student's t-test.

**Electrophoresis and immunoblotting**. Protein gel electrophoresis was performed by SDS-polyacrylamide gel electrophoresis (PAGE) using either home-made gels and Tris-glycine buffer or Novex 4–12% Bis-Tris gradient gels (Invitrogen) and MOPS buffer. Immunoblotting was performed using Hybond-C extra membranes (Amersham Biosciences) and a wet blot system (Invitrogen) and protein amounts were detected by Ponceau S staining (Sigma). After blocking the membrane with PBS containing 8% milk powder and 0.05% Tween 20, it was incubated with the primary antibodies as indicated. Uncropped blots are shown in Supplementary Fig. 8.

**Silverstain**. Proteins separated on a Novex 4–12% Bis-Tris gradient gel (Invitrogen) were fixed for 2 h in 50% methanol, 12% acetic acid, and 0.05% formalin, and the gel was washed three times with 35% ethanol for 20 min each. Afterwards, the gel was sensitized with 0.02% Na$_2$S$_2$O$_3$ and washed three times with MQ water for 5 min each. Proteins were stained with 0.2% AgNO$_3$ and 0.076% formalin for 20 min and the gel was washed again with MQ water. To develop the gel, it was incubated with 6% Na$_2$CO$_3$, 0.05% formalin, and 0.0004% Na$_2$S$_2$O$_3$ for approximately 5 min. The development was stopped by adding 50% methanol and 12% acetic acid for 5 min. The gel was stored in 1% acetic acid in the dark.

**Purification of His-SUMO2 conjugates**. Purification of His-SUMO2 conjugates was performed as follows[25]. Cells expressing His-SUMO-2 were washed and collected in ice-cold PBS. Small aliquots of cells were lysed in 1× LDS for total lysates samples. Guanidinium lysis buffer (6 M guanidinium-HCl, 0.1 M Na$_2$HPO$_4$/NaH$_2$PO$_4$, 0.01 M Tris/HCl, pH 8.0, and competing imidazole) was added to the cell pellet to lyse the cells, after which the cell lysates were sonicated to reduce the viscosity. These lysates were used to determine the protein concentration using the BCA Protein Assay Reagent (Thermo Scientific); lysates were equalized and His-SUMO-2 conjugates were enriched on nickel-nitrilotriacetic acid-agarose beads (Qiagen) after which the beads were washed using wash buffers A to D. Wash buffer A: 6 M guanidinium-HCl, 0.1 M Na$_2$HPO$_4$/NaH$_2$PO4, 0.01 M Tris/HCl, pH 8.0, 10 mM imidazole, pH 8.0, 5 mM β-mercaptoethanol, 0.2% Triton X-100. Wash buffer B: 8 M urea, 0.1 M Na$_2$HPO$_4$/NaH$_2$PO$_4$, 0.01 M Tris/HCl, pH 8.0, 10 mM imidazole, pH 8.0, 5 mM β-mercaptoethanol, 0.2% Triton X-100. Wash buffer C: 8 M urea, 0.1 M Na$_2$HPO$_4$/NaH$_2$PO$_4$, 0.01 M Tris/HCl, pH 6.3, 10 mM imidazole, pH 7.0, 5 mM β-mercaptoethanol, 0.2% Triton X-100. Wash buffer D: 8 M urea, 0.1 M Na$_2$HPO$_4$/NaH$_2$PO$_4$, 0.01 M Tris/HCl, pH 6.3, 5 mM β-mercaptoethanol, 0.1% Triton X-100. Samples were eluted in 7 M urea, 0.1 M Na$_2$HPO$_4$/NaH$_2$PO$_4$, 0.01 M Tris/HCl, pH 7.0, 500 mM imidazole.

**Crosslinking of antibody**. The CDC27 antibody was incubated overnight with Protein A beads at 4 °C. Prebleed serum was used as a negative control and incubated with Protein A beads in a similar manner. The beads were then washed two times with PBS and two times with 0.1 M sodium borate, pH 9 before incubating with 20 mM DMP (dimethyl pimelimidate dihydrochloride; D8388 Sigma) in 0.1 M sodium borate for 45 min at 4 °C. Incubation was repeated with fresh DMP solution for another 45 min at room temperature. Crosslinking was stopped by incubating twice for 10 min with 200 mM Tris pH 7.5 and 150 mM NaCl. The beads were washed twice with 50 mM glycine, pH 2.5 and 4 times with PBS plus 0.1% Tween 20 and stored as a 50% slurry in PBS plus 0.1% Tween 20 at 4 °C.

**Purification of endogenous APC/C**. CDC27 antibody (65 μg) was coupled to 50 μl Protein A beads as described above. One gram of HeLa cells was resuspended in 1 ml of lysis buffer (20 mM Tris pH 7.5, 150 mM NaCl, 2 mM EDTA, 0.05% Tween 20, 5% glycerol) and lysed using a dounce homogenizer. After centrifuging for 30 min at 20000 × g and 4 °C, the supernatant was incubated with CDC27 antibody coupled Protein A beads for 1 h at 4 °C. Beads were then washed four times with 20 mM Tris pH 7.5, 150 mM NaCl, 0.05% Tween 20, and 5% glycerol. The APC/C was eluted in elution buffer (20 mM Tris pH 7.5, 150 mM NaCl, 0.05% Tween 20, 2.5% glycerol, 1 mg ml$^{-1}$ CDC27 peptide).

**Purification of the APC/C via GFP-trap**. One gram of HeLa cells expressing GFP-APC WT and GFP-APC4 K772,798R mutant was lysed in a similar manner as described for the endogenous complex. Lysates were incubated with 20 μl GFP-trap (Chromotek) and incubated for 1 h at 4 °C. Beads were washed four times in 20 mM Tris pH 7.5, 150 mM NaCl, 0.05% Tween 20, and 5% glycerol and proteins were eluted in 2× NuPAGE LDS sample buffer (Thermo Fisher Scientific).

**Purification of recombinant proteins**. His10-tagged 3xSUMO2 was recombinantly expressed in BL21 cells by inducing protein expression at an OD$_{600}$ of 0.6 with 0.5 mM IPTG and incubating overnight at 25 °C. Cells were lysed in 50 mM HEPES pH7.6, 0.5 M NaCl, 25 mM MgCl$_2$, 20% glycerol, 0.1% N-P40, 50 mM imidazole, 1 mM phenylmethanesulfonylfluoride (PMSF), and protease inhibitor cocktail –EDTA (Roche), and the His-tagged 3xSUMO construct was purified from cells by incubating with Ni-NTA beads (Qiagen) for 3 h at 4 °C. Beads were then washed twice in lysis buffer including PMSF and protease inhibitor cocktail and twice in lysis buffer without PMSF and protease inhibitor cocktail. Proteins were eluted by incubating with lysis buffer plus 500 mM imidazole for 10 min at 4 °C. The elution step was repeated three times. Recombinant APC/C and Securin were kind gifts from Dr. B.A. Schulman (St. Jude Children's Research Hospital, Memphis, TN) and were purified[69,70]. Recombinant APC/C, containing a C-terminal Twin Strep-tag on APC4, was expressed in insect cells, and purified with a three-step scheme: (1) affinity purification with Strep-Tactin Sepharose (IBA Life

Sciences), (2) anion exchange chromatography, and (3) size exclusion chromatography (SEC). Securin (C197, 198A) was expressed in BL21(DE3) Codon Plus (RIL) cells as an N-terminal GST-TEV and C-terminal Cys-His6 fusion and purified by glutathione affinity chromatography, proteolytic TEV cleavage, nickel affinity chromatography, and SEC.

**In vitro SUMOylation of endogenous APC/C.** After immunoprecipitation, the endogenous APC/C was SUMOylated on the beads by incubating with 1.2 µg SAE1/2, 4 µg UBC9, 8 µg SUMO-3 or SUMO-2 allKR mutant, 50 mM Tris pH 7.5, 5 mM MgCl$_2$, 2 mM ATP, 3.5 U ml$^{-1}$ Creatine kinase, 10 mM Creatine phosphate, 0.6 U ml$^{-1}$ inorganic pyrophosphate, and protease inhibitor cocktail (Roche) in a 50 µl reaction for 3 h at 4 °C. A similar protocol was used to SUMOylate the APC/C exhibiting GFP-tagged APC4 bound to GFP-trap in vitro.

**Ubiquitylation assays with endogenous APC/C.** For in vitro ubiquitylation assays with endogenous APC/C, the SUMOylation machinery was removed by washing the beads three times with 20 mM Tris pH 7.5, 150 mM NaCl, 0.05% Tween 20 and 5% glycerol and 1 µg of CDH1 and 1.15 µg of an Hsl1 fragment (aa 667–872) were bound to the complex by incubating the beads for 1 h in 0.4% bovine serum albumin (BSA), 0.05% Tween 20, 50 mM Tris pH 7.5, and 150 mM NaCl. After washing another three times, the APC/C was eluted from the beads with 1 mg ml$^{-1}$ CDC27 peptide in wash buffer for 1 h (1:1 (w/v) ratio). Ubiquitylation of Hsl1 was performed by adding 40 nM UBE1 (Boston Biochem), 0.7 µM UbcH10, 8 µM Ubiquitin (Boston Biochem), 50 mM Tris pH 7.5, 5 mM MgCl$_2$, 2 mM ATP, and 1 mM dithiothreitol (DTT) and incubating at 4 °C.

**In vitro SUMOylation of recombinant APC/C.** To SUMOylate recombinant APC/C (kind gift of Dr. N. Brown and B.A. Schulman, Memphis, USA) in vitro, 1 µg of the complex was incubated with 160 ng SAE1/2, 2 µg UBC9, 2 µg SUMO2 allKR mutant, 50 mM Tris pH 7.5, 5 mM MgCl$_2$, 2 mM ATP, 3.5 U ml$^{-1}$ Creatine kinase, 10 mM Creatine phosphate, 0.6 U ml$^{-1}$ inorganic pyrophosphate, and protease inhibitor cocktail (Roche) for 3 h at 4 °C in a 15 µl reaction. As a negative control, recombinant APC/C WT was treated similarly but leaving out UBC9 from the reaction.

**Ubiquitylation assays with recombinant APC/C.** To obtain KIF18B as a substrate for the in vitro ubiquitylation reaction, five 15-cm dishes of U2OS cells were transfected with a pBIOBS-KIF18B-Flag construct. Three days after transfection the protein was purified from cells by immunoprecipitation. Cells were lysed in 1 ml of 20 mM Tris pH7.5, 150 mM NaCl, 2 mM EDTA, 0.05% Tween 20, 5% glycerol, sonicated, and centrifuged at 20,000 × g for 30 min at 4 °C. The supernatant was incubated with 30 µl of anti-Flag M2 affinity gel (Sigma) for 2 h at 4 °C, and the beads were washed four times with lysis buffer before eluting the protein with lysis buffer plus 1 mM Flag M2 epitope peptide for 30 min at 4 °C. Apart from KIF18B, recombinant Securin was used as a substrate in in vitro ubiquitylation assays.

For ubiquitylation, 1 µg of CDH1 was added to the SUMOylated APC/C and incubated for 30 min at 4 °C. Afterwards, 1 µg of substrate was ubiquitylated by adding 40 nM UBE1, 0.7 µM UbcH10, 8 µM Ubiquitin, 50 mM Tris pH7.5, 5 mM MgCl$_2$, 2 mM ATP, and 1 mM DTT in a 10 µl reaction and incubating at 4 °C for the indicated amount of time. One µl of 50 µM SENP2 catalytic domain (Boston Biochem) was added to the indicated samples after 2 h of ubiquitylation for an additional hour.

**Co-IP of binding partners of the APC/C.** Recombinant APC/C WT and APC/C mutant was SUMOylated in vitro by adding 50 mM Tris pH 7.5, 5 mM MgCl$_2$, 2 mM ATP, 3.5 U ml$^{-1}$ Creatine kinase, 10 mM Creatine phosphate, 0.6 U ml$^{-1}$ inorganic pyrophosphate, protease inhibitor cocktail (Roche), 6.4 µg SAE1/2, 40 µg UBC9, and 40 µg SUMO-3 to 10 µg samples of APC/C in a volume of 180 µl and incubating for 3 h at 4 °C. The SUMOylated recombinant complex was purified by adding Strep-tactin beads (GE Healthcare) and incubating for 2 h at 4 °C in 50 mM Tris pH 7.5 and 150 mM NaCl. Five 15-cm plates of HeLa cells per sample were lysed in 50 mM Tris pH 7.5, 150 mM NaCl, and 20 mM NEM (Sigma), sonicated to reduce viscosity and centrifuged at 20,000 × g for 30 min at 4 °C. The Strep-tactin beads were washed three times in lysis buffer to eliminate the SUMO machinery and incubated with the supernatant of the HeLa lysates for 2 h at 4 °C. Afterwards the beads were washed three times with 50 mM Tris pH 7.5 and 150 mM NaCl and three times with 50 mM ammonium bicarbonate. The bound proteins were incubated with 2 µg of trypsin on the beads overnight at 37 °C. The samples were then passed through a pre-washed 0.45 µm filter (Millipore) and acidified with 2% Trifluoroacetic acid (Sigma). For subsequent mass spectrometric analysis, the peptides were desalted by C-18 stage tips[36]. For western blot analysis, proteins were eluted from beads by incubating with 2xLDS.

**Co-IP of binding partners of 3xSUMO2.** Five 15-cm dishes of HeLa cells per sample were lysed in 1 ml of 50 mM Tris pH 7.5, 150 mM NaCl, 0.5% NP-40, 50 mM imidazole, sonicated and centrifuged at 20,000 × g for 1 h at 4 °C. The supernatant was incubated with recombinant 3xSUMO2 bound to Ni-NTA beads

for 2 h at 4 °C. As a control, HeLa lysate was incubated with Ni-NTA beads without 3xSUMO2. Samples were washed three times with 50 mM Tris pH 7.5, 150 mM NaCl, 0.5% NP-40, 50 mM imidazole and three times with 50 mM Tris pH 7.5, 150 mM NaCl including two changes of tubes. Binding partners of 3xSUMO2 were eluted by incubating with 8 M urea in 50 mM Tris pH 7.5 for 30 min at room temperature. A second elution was performed using 8 M urea, 50 mM Tris pH 7.5, and 500 mM imidazole for 30 min at room temperature.

**Binding of 3xSUMO2 to biotinylated peptides.** Peptides were synthesized on a SYRO II synthesizer, using preloaded Wang resin and standard Fmoc Solid Phase Peptide Chemistry, with PyBop and Dipea as activator and base and listed in Supplementary Table 2.

Liquid chromatography–tandem mass spectrometric (LC-MS) measurements for peptide synthesis were performed on a system equipped with a Waters 2795 Separation Module (Alliance HT), Waters 2996 Photodiode Array Detector (190–750 nm), Phenomenex Kinetex C18 (2.1× 50, 2.6 µm), and Xevo G2-XS Time of Flight Mass Spectrometer.

For the binding assay, wells of Streptavidin High Capacity Coated Plates (Sigma, 96-well, clear, S6940) were pre-washed three times with 200 µl 1×PBS per well. Peptides were added to the wells overnight at 4 °C at a concentration of 500 pmol ml$^{-1}$. Afterwards, blocking solution (0.4% BSA in 1×PBS) was added for 30 min at room temperature and the wells were washed three times with 200 µl 1×PBS/0.05%Tween 20. Then the wells were incubated with 50 µl of recombinant 3xSUMO2 (10 µg ml$^{-1}$) for 90 min at 4 °C. Unbound protein was washed away three times with 200 µl 1×PBS/0.05%Tween 20 and 50 µl of SUMO2/3 mouse monoclonal antibody (dilution 1:48) was added and incubated for 90 min at 4 °C. Wells were washed another three times with 200 µl 1×PBS/0.05%Tween 20 and 50 µl of horseradish peroxidase-coupled anti-mouse secondary antibody (dilution 1:200) was added and incubated for 90 min at 4 °C. Unbound antibody was washed away three times with 200 µl 1×PBS/0.05%Tween 20 and 100 ul of a 1:1 dilution of the reagents A and B in the Color Reagent Pack (R&D Systems) was added to the wells. The plate was incubated at room temperature in the dark until the positive controls were colored and the reaction was stopped with 50 µl 1 M H$_2$SO$_4$ per well. Binding of the peptides was determined by measuring the absorbance at 450 nm in a plate reader (Perkin Elmer Victor X3).

**Mass spectrometric analysis.** Desalted peptide samples were measured by nanoscale LC-MS/MS on an Orbitrap Q-Exactive (Thermo), and raw data analysis was performed using the MaxQuant Software and Perseus Software version 1.5.1.2.

**Data availability.** The authors confirm that all data supporting the findings of this study are available within the paper and its supplementary information files. The mass spectrometric proteomics data have been deposited to the ProteomeXchange Consortium via the PRIDE partner repository with the dataset identifier PXD006335.

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

## Acknowledgements

The authors are grateful to Dr. B.A. Schulman, Dr. N. Brown, Dr. J.M. Peters, Dr. J. Nilsson, Dr. R.T. Hay, Dr. P. de Graaf, and Dr. H.T. Timmers for reagents and advice.

The authors are also grateful to Dr. M. Matunis for sharing unpublished data. The laboratory of A.C.O.V. is supported by the European Research Council (ERC) and the Netherlands Organisation for Scientific Research (NWO).

## Author contributions

K.E., S.A.G.C., E.W., and J.A.R. performed experiments. K.E. and A.C.O.V. wrote the manuscript. All authors commented on the manuscript. R.H.M. provided reagents, expertise, and feedback. D.E.A and H.O. provided peptides. A.C.O.V. conceived the project and secured funding.

## Additional information

**Competing interests:** The authors declare no competing interests.

