## [Peer Review File(PDF 617 kb) · Nature Communications]

Reviewer #1 (Remarks to the Author):

The manuscript by Eilfer et al explores the consequences of APC/C SUMOylation on the metaphase to anaphase transition and APC/C activity.

They show that inhibition of SUMOylation through knockdown of the SUMO activating enzyme subunits SAE1 and SAE2 delayed mitotic progression and resulted in the formation of chromatin bridges. They confirm that APC/C is SUMOylated and that Lys 772 and Lys 798 are the major sites of SUMOylation. Interestingly the authors show that Lys 772 SUMOylation is stimulated by CK1 phosphorylation of the neighboring Ser residues (777 and 779). SUMOylated APC/C has a higher activity towards the model substrate Hsl1.

The SUMOylated APC/C enhances binding to KIF18B, and this is associated with increased ubiquitylation of KIF18B. Since KIF18B interacts with SUMO and contains a SIM, it is suggested that enhanced APC/C-Kif18B interactions mediated by SUMO moieties attached to Apc4 and the SIM motif on KIF18B result in stimulated KIF18B ubiquitylation.

The findings in this manuscript that APC/C SUMOylation stimulates its activity to specific substrates such as KIF18B through enhanced affinity for substrate, mediated by SUMO sites on Apc4 are important and interesting for the SUMO and cell cycle fields.

Major Points.

1. The authors show that ubiquitination of both Hsl1 and Kif18B is promoted by APC/C SUMOylation and propose that SIM motifs in Hsl1 and KIF18B mediate interactions with SUMO moieties on the APC/C. This proposal could be tested for at least one of these substrates by mutating the SIM motif.
2. The suggestion that SUMOylation of the APC/C stimulates Kif18B ubiquitination and degradation at metaphase could be tested using their SAE1 and SAE2 knock-down cell lines. According to their model, Kif18B, but not securin, should be stabilized when APC/C SUMOylation is inhibited.
3. Ubiquitylation of securin is not regulated by APC/C SUMOylation. Did the authors investigate other substrates and specifically which if any other APC/C substrates contain the SIM, and if so whether their ubiquitylation is regulated by APC/C ubiquitylation?

Reviewer #2 (Remarks to the Author):

In this manuscript Eifler et al. describe a delay at the transition from meta- to anaphase after knockdown of the essential SUMO E1 enzyme in mammalian cells; as follow-up they focus on investigating the regulation of the APC/C by sumoylation. They identify two lysine residues in APC4 to be modified with Sumo in human cells, and demonstrate that modification of one of these sites can be regulated by phosphorylation. They show that sumoylation of APC4 impacts on APC/C ubiquitylation activity towards a subset of targets, e.g., the model substrate Hsl but not the well-known APC/C substrate Securin. Moreover, in a screen for substrates specific for the sumoylated form of the APC/C they identify KIF18B as a potential novel substrate of the APC/C, and observe that indeed KIF18B preferentially associates with and is ubiquitylated by the sumoylated form of the APC/C in vitro.

Both, sumoylation and the APC/C are important for the regulation of mitotic progression; so far however, our understanding of how sumoylation contributes to mitotic regulation is still limited. Overall, this work is a technically sound study that provides novel and interesting insights into how sumoylation contributes to APC/C regulation, and would be adequate for publication in Nature

Communications. However, there are still some points of criticism, which should be addressed before the manuscript is suitable for publication. In addition, the study would certainly benefit from additional insights into how APC4 sumoylation influences substrate ubiquitylation, or in vivo evidence that APC4 sumoylation influences mitotic progression or KIF18B. All points of criticism are listed in detail below.

Major comments:

1. Is it possible that all supplemental tables are missing? at least I did not find them...
2. The authors suggest that no other APC/C subunit besides APC4 is sumoylated to significant levels, based on a SUMO2/3 immunoblot of purified wt vs mut GFP-APC4 complexes (p.10, l.301 + Fig. 3C). The authors state that they can detect the other APC/C subunits by mass spec (missing suppl. table 1), however mass spec is generally much more sensitive than immunoblotting and does not allow much conclusion as to relative protein abundance. A Coomassie or silver stained SDS PAGE of the purified GFP-APC/C complexes would at least provide some idea for relative protein abundances of APC4 in comparison to the other APC/C subunits.
3. Based on an in vitro sumoylation assay in the absence or presence of CK2 (Fig. 4B), the authors conclude that phosphorylation of an APC4 fragment enhances its sumoylation (p.11, l. 326). However, in this experiment they do not show any evidence for phosphorylation (e.g. could also be an effect of CK2 on the sumoylation machinery); to address this issue the authors should include a phospho-deficient APC4 mutant in this experiment.
4. In order to evaluate whether there are differences in the levels of associated KIF18B (Fig. 7A) equal levels of bound WT and mutant APC/C are absolutely crucial. Here, the authors need to provide immunoblots for APC4 and one other APC/C subunit.
5. The authors test binding of KIF18B to His-SUMO2 chains in a His-pulldown experiment (Fig. 7B) and conclude that KIF18B strongly binds to SUMO2 chains (p. 14). I am confused with the experimental setup. According to the Figure legend 7B and Methods section (p. 31) KIF18B binding seems to be largely resistant to elution with 8M urea and only elute upon addition of imidazole – is it possible that accidentally there is a mistake in the experimental description? 8M urea is denaturing and will certainly disrupt SUMO – SIM interactions (KD typically in the range of a few μ M), which the authors suggest to contribute to the KIF18B-APC/C interaction (p.17). I have no idea which type of interaction except for a covalent one or a direct interaction of histidine residues with the Ni-NTA matrix would resist 8M urea. This issue needs clarification.
6. The title of this manuscript claims that “Sumo targets the APC/C [...] to regulate transition from metaphase to anaphase”. This idea is certainly supported by the fact that the APC/C is generally known to be an important mitotic regulator and by a study by the Matunis lab showing that sumoylation of APC4 is upregulated in mitosis (also mentioned in the discussion). However, the in vivo evidence provided in this study is rather scarce. None of the experiments addressing sumoylation of the APC/C has been performed from a mitotic cell population and there is also no experiment showing that KIF18B is a (mitotic) target of the APC/C in cells. This could be addressed by replacing the endogenous APC4 with a wt vs sumoylation-deficient variant in cells to subsequently analyze its effect on KIF18B ubiquitylation.

Other comments:

1. The knowledge that APC2 is not regulated by neddylation due to the lack of the neddylation site has been reported long ago and is not a novel finding by the authors (e.g. see review by Pan et al., *Oncogene*, 2004); this should be acknowledged in the way the authors present their experiment and by citation. Likewise, it has previously been reported that knockdown of the sumoylation machinery leads to defects in chromosome segregation and DNA bridges (e.g. Nacerddine et al., *Dev. Cell*, 2005; He et al., *Plos One*, 2015).
2. In Fig. 5A, a silver-stained gel of sumoylated and non-sumoylated APC/C is shown to confirm the presence of all APC/C subunits. Again, I could not find the supporting suppl. table 2 with the MS data; in addition however, I am wondering why the APC2-labeled band is not visible after sumoylation, if no APC subunit besides APC4 is sumoylated?
3. The immunoblots shown in Figure 5 are rather saturated. Nevertheless, the authors make a quantitative statement that at least one third of the APC4 is modified with two Sumos (p.13,

l.369). This quantitative conclusion is not possible; otherwise I would have to conclude that there was three times as much APC4 in the reaction +Sumo compared to -Sumo (Fig. 5C). Similarly, the APC11 blot in Fig. 5D is not suitable as loading control – could the authors eventually provide a non-saturated exposure?

4. In figure legend 7C, Senp2 is not being mentioned. This may lead to confusion as the figure does not allow any conclusion as to when Senp2 has been added and therefore some reader may wonder why Senp2 does not affect KIF18B ubiquitylation.

Reviewer #3 (Remarks to the Author):

Eifler et al investigate the regulatory function of SUMOylation on mitotic progression. RNAi depletion of SAE1 or SAE2 caused mitotic defects including delayed anaphase onset and defects in chromosome segregation. Previous studies identified APC4 to be SUMO modified. Eifler et al can confirm this result and identify K772 and K798 of APC4 as SUMO acceptor residues. Furthermore, the authors provide evidence that K772 SUMOylation is regulated by S777 and S779 phosphorylation. In vitro assays revealed that SUMOylation of APC4 increases ligase activity of the APC towards Kif18B. The study presents an interesting finding. However, there are several significant concerns regarding the quality of the presented data and therefore this reviewer cannot recommend publication of the submitted ms in Nature Communication.

Specific comments:

Fig. 1: This figure lacks several important controls. No western blot is provided to demonstrate that SAE1 or SAE2 are indeed depleted. No rescue experiment has been performed to prove that the observed phenotype is indeed due to the depletion of SAE1 or SAE2. This is standard and should be included with any RNAi experiment.

Knockdown of SAE2 with ishSAE2.1 leads to dramatic increase in the formation of chromosome bridges compared to knockdown with ishSAE2.2. Yet, both shRNAs have almost identical effects on mitotic timing (1B). This does not make sense.

Fig. 4. B This experiment lacks the important control of the S777/779A mutant. This is an essential control.

Fig. 4.C The authors argue that mutating S777 and S779 to alanine resulted in decreased SUMOylation at K772. From the WB it is difficult to judge if the ratio of unmodified to modified APC4 is indeed affected by the S777/779A mutation. Why is there no difference in the level of SUMOylation of wt APC4 compared to the phosphomimic mutant? Is casein kinase II believed to be constitutively active?

Fig. 4.D Importantly: Why does the treatment with quinalizarin equally affect the mono- and di-SUMOylation. According to the authors, only the SUMOylation of K772 should be affected.

Fig. 5.B and C. According to the WB, a substantial fraction of APC4 is unmodified after in vitro SUMOylation. Yet, according to the silverstain gel the unmodified version of APC4 completely disappears. What happens to APC2 upon in vitro SUMOylation?

Fig. 5.D Importantly: The authors argue that equal levels of the APC were present during the reactions. However, APC11 WB in the - SUMO reaction clearly indicates uneven levels with the lowest amount being present in the later time points.

In the discussion, the authors discuss how altered kinetics in the destruction of distinct APC substrates could account for the observed phenotypes. This reviewer is puzzled by the provided

explanations. How should e.g. altered kinetics of Kif18B destruction account for the observed delay in anaphase onset? Anaphase onset is triggered by silencing the mitotic checkpoint resulting in APC activation. So how could the impaired destruction of Kif18B in SAE1/SAE2 depleted cells delay anaphase onset if one has to assume that destruction of Kif18B does not take place until the mitotic checkpoint is silenced? Do the authors propose that Kif18B is degraded under active checkpoint conditions?

Reviewer #1 (Remarks to the Author):

The manuscript by Eilfer et al explores the consequences of APC/C SUMOylation on the metaphase to anaphase transition and APC/C activity. They show that inhibition of SUMOylation through knockdown of the SUMO activating enzyme subunits SAE1 and SAE2 delayed mitotic progression and resulted in the formation of chromatin bridges. They confirm that APC/C is SUMOylated and that Lys 772 and Lys 798 are the major sites of SUMOylation. Interestingly the authors show that Lys 772 SUMOylation is stimulated by CK1 phosphorylation of the neighboring Ser residues (777 and 779). SUMOylated APC/C has a higher activity towards the model substrate Hsl1.

The SUMOylated APC/C enhances binding to KIF18B, and this is associated with increased ubiquitylation of KIF18B. Since KIF18B interacts with SUMO and contains a SIM, it is suggested that enhanced APC/C-Kif18B interactions mediated by SUMO moieties attached to Apc4 and the SIM motif on KIF18B result in stimulated KIF18B ubiquitylation.

The findings in this manuscript that APC/C SUMOylation stimulates its activity to specific substrates such as KIF18B through enhanced affinity for substrate, mediated by SUMO sites on Apc4 are important and interesting for the SUMO and cell cycle fields.

In reply: We are grateful for the support of the reviewer concerning overall relevance of our manuscript.

Major Points.

1. The authors show that ubiquitination of both Hsl1 and Kif18B is promoted by APC/C SUMOylation and propose that SIM motifs in Hsl1 and KIF18B mediate interactions with SUMO moieties on the APC/C. This proposal could be tested for at least one of these substrates by mutating the SIM motif.

In reply: We have tested the hypothesis that KIF18B harbours SIMs in the following manner. First, we have analyzed the KIF18B sequence to locate potential SIM motifs. This yielded 12 potential SIMs as detailed below. Next, we have synthesized peptides containing these SIMs and flanking amino acids plus

controls where the large hydrophobic amino acids were replaced for alanines, to block interaction with SUMO according to common practice in the field. These peptides contained biotin and a PEG linker to enable binding to Avidin coated 96-well plates.

The synthesized set of peptides was as follows:

KIF18B 12-26	(Biotin)VEDSTLQVVVRVRPP	WT
KIF18B 12-26	(Biotin)VEDSTAQAAARVRPP	mutant
KIF18B 33-47	(Biotin)SQRRPVVQVVDERVL	WT
KIF18B 33-47	(Biotin)SQRRPAAQAADERVL	mutant
KIF18B 41-54	(Biotin)VVDERVLVFNPEEP	WT
KIF18B 41-54	(Biotin)VVDERAAAFNPEEP	mutant
KIF18B 167-181	(Biotin)VYNEQIHDLLEPKGP	WT
KIF18B 167-181	(Biotin)VYNEQAHDAAEPKGP	mutant
KIF18B 186-199	(Biotin)EDPDKGVVVQGLSF	WT
KIF18B 186-199	(Biotin)EDPDKGAAAQGLSF	mutant
KIF18B 203-217	(Biotin)ASAEQLLEILTRGNR	WT
KIF18B 203-217	(Biotin)ASAEQAAEAATRGNR	mutant
KIF18B 256-269	(Biotin)VAKMSLIDLAGSER	WT
KIF18B 256-269	(Biotin)VAKMSAADAAGSER	mutant
KIF18B 285-299	(Biotin)NINRSLALINVLNA	WT
KIF18B 285-299	(Biotin)NINRSAAAAANVLNA	mutant
KIF18B 315-328	(Biotin)SKLTRLLKDSLGGN	WT
KIF18B 315-328	(Biotin)SKLTRAAKDSLGGN	mutant
KIF18B 355-369	(Biotin)DRAKEIRLSLKSNT	WT
KIF18B 355-369	(Biotin)DRAKEARASAKSNVT	mutant
KIF18B 525-538	(Biotin)SKRLALKVLCVAQR	WT
KIF18B 525-538	(Biotin)SKRLAAKAACVAQR	mutant
KIF18B 821-836	(Biotin)RPAGPLVLPPLSPL	WT
KIF18B 821-836	(Biotin)RPAGPAAAPLPLSPL	mutant
RNF4 41-54	(Biotin)TAGDEIVDLTCESL	WT
RNF4 41-54	(Biotin)TAGDEAADATCESL	mutant
RNF4 53-66	(Biotin)SLEPVVVDLTHNDS	WT
RNF4 53-66	(Biotin)SLEPVAADATHNDS	mutant

Next, we incubated these peptides after binding on the 96-well plates with SUMO polymers to verify SUMO binding and performed an anti SUMO ELISA. As positive controls, we included two high affinity SIMs derived from the well-known SUMO binder RNF4. This resulted in the confirmation of the SIM LLALI in KIF18B (aa 290-294) as a SUMO binder at about 20% affinity compared to the super SIMs in RNF4 (Figure 8C). However, this SIM cannot be the only SUMO binding site in KIF18B, since mutating this SIM in the context of the full size protein did not reduce its binding to SUMO chains.

2. The suggestion that SUMOylation of the APC/C stimulates Kif18B ubiquitination and degradation at metaphase could be tested using their SAE1 and SAE2 knock-down cell lines. According to their model, Kif18B, but not securin, should be stabilized when APC/C SUMOylation is inhibited.

In reply: We verified the ubiquitination of KIF18B in the phosphosite plus PTM database [<http://www.phosphosite.org>]. Human KIF18B is indeed ubiquitinated and the ubiquitin acceptor sites are lysines 62, 304, 688, 783, 795 and 796. We tested KIF18B ubiquitination using His-ubiquitin enrichment and anti KIF18B immunoblots, but did not get a proper KIF18B signal. We hypothesize that this could potentially be due to epitope masking by ubiquitination. Additionally, ubiquitylation might be restricted in time. Given these practical challenges, we were not able to fully address this point of the reviewer.

3. Ubiquitylation of securin is not regulated by APC/C SUMOylation. Did the authors investigate other substrates and specifically which if any other APC/C substrates contain the SIM, and if so whether their ubiquitylation is regulated by APC/C ubiquitylation?

In reply: We have verified whether other APC/C substrates have SUMO Interaction Motifs. In total, we found 77 potential APC/C substrates in the literature. In this set, we found 35 proteins that contained one or more SUMO Interaction Motifs. The details are shown in the Supplementary table 4 and are as follows:

Substrate	UniProt ID	SUMO interaction motifs
Anillin ¹	Q9NQW6	DPKVEQK IEVI REIEMSVD (544 – 547) DAANYYY LII LKAGAENMV (833 – 836) ARRNTF ELITV RPQREDDR (1058 – 1061) MQKLNQV LVDI RLWQPDAC (1106 – 1109)
Aurora A ²	O14965	None found
Aurora B ³	Q96GD4	None found
B99 ⁴	Q9NYZ3	DDPKKED ILL LADKDFDFD (9 – 12) NRKTDSR LVDV SPDRGSP (571 – 575) VGSESR LIDL MTNTPDMN (670 – 673)
BARD1 ⁵	Q99728	GQRRDGP LVL IGSGLSSEQ (570 – 573) DCVMSF ELLPLD S***** (772 – 776)
Bub1 ⁶	O43683	ESNMERR VITIS KSEYSVH (211 – 215) IRALRN LIVLL LECKRSR (1073 – 1077)
Cdc6 ⁷	Q99741	TAEKGPM IVLVL DEMDQLD (279 – 283) LSNSHLV LIGI ANTLDLTD (313 – 316) LDVCRRA IEIVE SDVKSQT (398 – 402)

Cdc20 ⁸	Q12834	None found
Cdh1 ⁹	Q9UM11	ERRLLRQ IVI QNTMPRV (13 - 15) TKVKWES VSVL NLFTRIR* (486 - 489)
CDR2 ¹⁰	Q01850	None found
CENPF ¹¹	P49454	None found
Centrin ¹²	O15182	None found
Cdc25A ¹³	P30304	VPTDGKR VIVV FHCEFSSE (425 - 428)
CDT1 ¹⁴	Q9H211	None found
CKAP2 ¹⁵	Q8WWK9	ITSPIEN IIAI YEKAILAG (480 - 483) IEEMRHT IVDIL TMKSQEK (502 - 506)
CKS1 ¹⁶	P61024	None found
Claspin ¹⁷	Q9HAW4	None found
Cyclin A2 ¹⁸	P20248	None found
Cyclin B1 ¹⁹	P14635	PLEKVPML VVPV SEPVPE (89 - 92) EKLSPEP ILVDT ASPSPME (120 - 124)
Cyclin B3 ²⁰	Q8WWL7	NIMEKPL ILDIST TTSKTPN (136 - 140) CDCEAQQ LVL ***** (1393 - 1395)
DRP1 ²¹	O00429	DI IQLPQ IVVVG TQSSGKS (28 - 31) FPKLHDA IVEVV TCLLRKR (464 - 468)
E2C ²²	O00762	None found
E2F1 ²³	Q01094	RLLDSSQ IVII SAAQDASA (33 - 37) LSHSADG VVDLN WAAEVLK (150 - 153)
E2F3 ²⁴	O00716	SGLKDQT VIVV KAPPETRL (315 - 318)
ECT2 ²⁵	Q9H8V3	None found
FoxM1 ²⁶	Q08050	None found
G9A ²⁷	Q96KQ7	WAAEHKH IEVI RMLLTRGA (798 - 801)
Geminin ²⁸	O75496	None found
GLP1 ²⁷	Q9H9B1	None found
Glutaminase 1 ²⁹	O94925	None found
HEC1 ³⁰	O14777	None found
Hmmr ⁵	O75330	KQSLEEN IVILS KQVEDLN (279 - 283)
HSF2 ³¹	Q03933	NMYGFRK VVHID SGIVKQE (73 - 77) RERISDD IIIYD VTDNDAD (246 - 250) NCSQYPD IVIVE DDNEDEY (284 - 288)
HURP ⁵	Q15398	PTLEGRI LVELD ETSQGLV (60 - 64)
ID2 ³²	Q02363	LQHVIDY ILD LQIALDSHP (72 - 75) ALDSHPT IVSL HHRPGQN (85 - 88)
JNK1 ³³	P45983	None found
JNK2 ³³	P45984	None found
KID ³⁴	Q14807	ESFLKAN ILGL AAGQRCGA (653 - 656)
KIFC1 ³⁵	Q9BW19	None found
KIF18A ³⁵	Q8NI77	None found
KIF2C ³⁵	Q99661	RMHGKFS LVDL AGNERGAD (490 - 493)
KIF4A ³⁵	O95239	LEIYNEE ILD LLCPSREKA (148 - 152) DQELKEN VEII CNLQQLIT (463 - 466)
MCL1 ³⁶	Q07820	DELYRQS LEII SRYLREQA (179 - 183)
MFN1 ³⁷	Q8IWA4	CALLRDD LVLVD SPGTDVT (174 - 178) TSRTSMG IIIV GGVIWKT (599 - 602)
MOAP-1 ³⁸	Q96BY2	None found

MPS1 ³⁹	P33981	None found
NEK2A ⁴⁰	P51955	None found
NIPA ⁴¹	Q86WB0	None found
NLP ⁴²	Q9Y2I6	HDAQRKEIEVLKDKKAC (1121 - 1124)
NuSAP ⁵	Q9BXS6	VLGMRRLILIAED***** (436 - 440)
OCT1 ⁴³	P14859	None found
OPA1 ³⁷	O60313	LIDMYSEVLDVLSDYDASY (271 - 275) TQDHLPRVVVGDQSAGKT (291 - 294)
p21 ⁴⁴	P38936	None found
p63 ⁴⁵	Q9H3D4	GMNRRPILIIIVTLETRDGO (322 - 326)
p190RhoGAP ⁴⁶	Q9NRY4	LAKTKKPIVVVLTKCDEGV (195 - 199) SDPNIDRINLVILGKDGLA (598 - 602) FLCEVQDIIPIQLVALTDG (880 - 883) HDSTQGKIIITIRNINKAQS (1116 - 1119) ATRTYQTIIELEFIQQCPFF (1424 - 1427)
PAF15 ⁴⁷	Q15004	None found
PFKFB3 ⁴⁸	Q16875	LTNSPTVIVMVGLPARGKT (38 - 41) RDLSLIKVIDVGRFLVNR (215 - 218)
PIF1 ⁴⁹	Q9H611	None found
PLK1 ⁵⁰	P53350	None found
RacGAP1 ⁵¹	Q9H0H5	TSPMIPSIVVHCVNEIEQR (365 - 368)
RAP80 ⁵²	Q96RL1	None found
RASSF1A ⁵³	Q9NS23	None found
RCS1 ⁵⁴	Q9BSJ6	None found
Securin ⁵⁵	O95997	None found
SGO1 ⁵⁶	Q5FBB7	None found
SKP2 ¹⁶	Q13309	None found
SnoN ⁵⁷	P12757	None found
Sororin ⁵⁸	Q96FF9	EAAEQFDLLVE***** (249 - 252)
Sp100 ⁵⁹	P23497	HHNQASDIIVISSESEGS (323 - 327)
TFAM ³⁷	Q00059	None found
TK1 ⁶⁰	P04183	None found
TMPK ⁶¹	P23919	MAARRGALIVLEGVDRAGK (8 - 12)
TOME-1 ⁶²	Q99618	None found
TRB3 ⁶³	Q96RU7	EEEGDREVVLVYG***** (354 - 357)
TPX2 ⁶⁴	Q9ULW0	None found
USP1 ⁶⁵	O94782	None found
USP37 ⁶⁶	Q86T82	EEELLAAVLEISKRDASPS (712 - 716)

Time constrains did not permit further testing of these substrates.

Reviewer #2 (Remarks to the Author):

In this manuscript Eifler et al. describe a delay at the transition from meta- to anaphase after knockdown of the essential SUMO E1 enzyme in mammalian cells; as follow-up they focus on investigating the regulation of the APC/C by sumoylation. They identify two lysine residues in APC4 to be modified with Sumo in human cells, and demonstrate that modification of one of these sites can be regulated by phosphorylation. They show that sumoylation of APC4 impacts on APC/C ubiquitylation activity towards a subset of targets, e.g., the model substrate Hsl but not the well-known APC/C substrate Securin. Moreover, in a screen for substrates specific for the sumoylated form of the APC/C they identify KIF18B as a potential novel substrate of the APC/C, and observe that indeed KIF18B preferentially associates with and is ubiquitylated by the sumoylated form of the APC/C in vitro.

Both, sumoylation and the APC/C are important for the regulation of mitotic progression; so far however, our understanding of how sumoylation contributes to mitotic regulation is still limited. Overall, this work is a technically sound study that provides novel and interesting insights into how sumoylation contributes to APC/C regulation, and would be adequate for publication in Nature Communications. However, there are still some points of criticism, which should be addressed before the manuscript is suitable for publication. In addition, the study would certainly benefit from additional insights into how APC4 sumoylation influences substrate ubiquitylation, or in vivo evidence that APC4 sumoylation influences mitotic progression or KIF18B. All points of criticism are listed in detail below.

In reply: We are grateful for the support of the reviewer concerning overall relevance of our manuscript.

Major comments:

1. Is it possible that all supplemental tables are missing? at least I did not find them...

In reply: We have double checked that the supplementary tables are included in the resubmission.

2. The authors suggest that no other APC/C subunit besides APC4 is sumoylated to significant levels, based on a SUMO2/3 immunoblot of purified wt vs mut GFP-APC4 complexes (p.10, l.301 + Fig. 3C). The authors state that they can detect the other APC/C subunits by mass spec (missing suppl. table 1), however mass spec is generally much more sensitive than immunoblotting and does not allow much conclusion as to relative protein abundance. A Coomassie or silver stained SDS PAGE of the purified GFP-APC/C complexes would at least provide some idea for relative protein abundances of APC4 in comparison to the other APC/C subunits.

In reply: We have included a silverstained gel of the purified GFP-APC/C complexes in supplementary Figure 3B. The overall purity of the GFP-APC/C complexes is not as good as for the endogenous complexes as shown in Figure 6B. Therefore, we have verified equal purification levels of APC3 and APC7 as included in Supplementary Figure 3C.

3. Based on an in vitro sumoylation assay in the absence or presence of CK2 (Fig. 4B), the authors conclude that phosphorylation of an APC4 fragment enhances its sumoylation (p.11, l. 326). However, in this experiment they do not show any evidence for phosphorylation (e.g. could also be an effect of CK2 on the sumoylation machinery); to address this issue the authors should include a phospho-deficient APC4 mutant in this experiment.

In reply: We have generated a recombinant C-terminal fragment of APC4 where serines 772 and 779 were replaced for alanines to avoid phosphorylation of both sites. We compared this mutant to wild-type APC4 concerning CK2 phosphorylation and subsequent SUMOylation. Unfortunately, no difference was observed in SUMOylation (Supplementary Figure 5). This could be either due to an effect of CK2 on the SUMOylation machinery as mentioned by the reviewer, or could be due to additional phosphorylation of APC4 at adjacent residues including serines 757, 758, 765 and 808. Given these results, we have de-emphasized the role of phosphorylation in the manuscript.

4. In order to evaluate whether there are differences in the levels of associated KIF18B (Fig. 7A) equal levels of bound WT and mutant APC/C are absolutely crucial. Here, the authors need to provide immunoblots for APC4 and one other APC/C subunit.

In reply: We have added APC7 as loading control (Supplementary Figure 7)

5. The authors test binding of KIF18B to His-SUMO2 chains in a His-pulldown experiment (Fig. 7B) and conclude that KIF18B strongly binds to SUMO2 chains (p. 14). I am confused with the experimental setup. According to the Figure legend 7B and Methods section (p. 31) KIF18B binding seems to be largely resistant to elution with 8M urea and only elute upon addition of imidazole – is it possible that accidentally there is a mistake in the experimental description? 8M urea is denaturing and will certainly disrupt SUMO – SIM interactions (KD typically in the range of a few μM), which the authors suggest to contribute to the KIF18B-APC/C interaction (p.17). I have no idea which type of interaction except for a covalent one or a direct interaction of histidine residues with the Ni-NTA matrix would resist 8M urea. This issue needs clarification.

In reply: We do see KIF18B already coming up with the first elution (8M urea) showing that it is not a covalent modification of KIF18B with SUMO but that it is binding of KIF18B to SUMO in a non-covalent fashion. However, using a relatively small elution volume and fluid remaining between the beads might be responsible for the considerable amount of KIF18B present in the second elution. Given the size of the band, we can rule out covalent interaction between the SUMO trimer and KIF18B.

6. The title of this manuscript claims that “Sumo targets the APC/C [...] to regulate transition from metaphase to anaphase”. This idea is certainly supported by the fact that the APC/C is generally known to be an important mitotic regulator and by a study by the Matunis lab showing that sumoylation of APC4 is upregulated in mitosis (also mentioned in the discussion). However, the in vivo evidence provided in this study is rather scarce. None of the experiments addressing sumoylation of the APC/C has been performed from a mitotic cell population and there is also no experiment showing that KIF18B is a (mitotic) target of the APC/C in cells. This could be addressed by replacing the endogenous APC4 with a wt vs sumoylation-deficient variant in cells to subsequently analyze its effect on KIF18B ubiquitylation.

In reply: this is indeed the key point to strengthen. We have knocked down endogenous APC4 and performed rescue experiments using wild-type APC4 and the APC4 K772, 798R double mutant resistant against the knockdown and measured the time required for transition from metaphase to anaphase in live-cell microscopy experiments (Figure 4). As expected, knockdown of APC4 resulted in a delay in metaphase to anaphase transition. Excitingly, the results also showed that whereas this effect can

efficiently be rescued using wild-type APC4, it can't be rescued using the K772, 798R double mutant. This provides important evidence for the functional relevance of APC4 SUMOylation.

Other comments:

1. The knowledge that APC2 is not regulated by neddylation due to the lack of the neddylation site has been reported long ago and is not a novel finding by the authors (e.g. see review by Pan et al., Oncogene, 2004); this should be acknowledged in the way the authors present their experiment and by citation. Likewise, it has previously been reported that knockdown of the sumoylation machinery leads to defects in chromosome segregation and DNA bridges (e.g. Nacerddine et al., Dev. Cell, 2005; He et al., Plos One, 2015).

In reply: we have included these citations as requested.

2. In Fig. 5A, a silver-stained gel of sumoylated and non-sumoylated APC/C is shown to confirm the presence of all APC/C subunits. Again, I could not find the supporting suppl. table 2 with the MS data; in addition however, I am wondering why the APC2-labeled band is not visible after sumoylation, if no APC subunit besides APC4 is sumoylated?

In reply: We have double checked the inclusion of Supplementary table 2 in the resubmission. This table confirms that all APC/C subunits are present, but is not informative concerning overall levels, since quantitation cannot be properly performed on single samples. In addition, we have now included in Figure 6D a confirmation of equal APC2 and APC11 levels, since APC11 is the RING subunit of the APC/C complex.

3. The immunoblots shown in Figure 5 are rather saturated. Nevertheless, the authors make a quantitative statement that at least one third of the APC4 is modified with two Sumos (p.13, l.369). This quantitative conclusion is not possible; otherwise I would have to conclude that there was three times as much APC4 in the reaction +Sumo compared to -Sumo (Fig. 5C). Similarly, the APC11 blot in Fig. 5D is not suitable as loading control – could the authors eventually provide a non-saturated exposure?

In reply: non-saturated exposures of the APC4 blot in Figure 6C and of the APC11 blot in Figure 6D are now included.

4. In figure legend 7C, Senp2 is not being mentioned. This may lead to confusion as the figure does not allow any conclusion as to when Senp2 has been added and therefore some reader may wonder why Senp2 does not affect KIF18B ubiquitylation.

In reply: we have now added a description on the timing of the addition of Senp2 to the legend of new Figure 8D to avoid confusion.

Reviewer #3 (Remarks to the Author):

Eifler et al investigate the regulatory function of SUMOylation on mitotic progression. RNAi depletion of SAE1 or SAE2 caused mitotic defects including delayed anaphase onset and defects in chromosome segregation. Previous studies identified APC4 to be SUMO modified. Eifler et al can confirm this result and identify K772 and K798 of APC4 as SUMO acceptor residues. Furthermore, the authors provide evidence that K772 SUMOylation is regulated by S777 and S779 phosphorylation. In vitro assays revealed that SUMOylation of APC4 increases ligase activity of the APC towards Kif18B. The study presents an interesting finding. However, there are several significant concerns regarding the quality of the presented data and therefore this reviewer cannot recommend publication of the submitted ms in Nature Communication.

In reply: We are grateful for the support of the reviewer concerning overall interest in our findings.

Specific comments:

Fig. 1: This figure lacks several important controls. No western blot is provided to demonstrate that SAE1 or SAE2 are indeed depleted. No rescue experiment has been performed to prove that the observed phenotype is indeed due to the depletion of SAE1 or SAE2. This is standard and should be included with any RNAi experiment.

Knockdown of SAE2 with ishSAE2.1 leads to dramatic increase in the formation of chromosome bridges compared to knockdown with ishSAE2.2. Yet, both shRNAs have almost identical effects on mitotic timing (1B). This does not make sense.

In reply: We have now verified the SAE2 knockdown in Figure 1C. In line with the reduction in SUMO conjugates as shown in the first panel of this figure, we observed an excellent knockdown efficiency of both SAE2 knockdown constructs. Moreover, since SAE1/SAE2 forms a dimer, we observed a reduction in SAE2 levels upon SAE1 knockdown which could represent the interdependence of these subunits for overall dimer stability. We have not found a proper antibody to address SAE1 levels, but confirmed the reduction in SUMO conjugates in the first panel of the figure. Given that we see significant effects on metaphase to anaphase transition for all three independent knockdown constructs, we consider it highly unlikely that these are off target effects.

Fig. 4. B This experiment lacks the important control of the S777/779A mutant. This is an essential control.

In reply: We agree that this is an essential control indeed. We have generated a recombinant C-terminal fragment of APC4 where serines 772 and 779 were replaced for alanines to avoid phosphorylation of both sites. We compared this mutant to wild-type APC4 concerning CK2 phosphorylation and subsequent SUMOylation. Unfortunately, no difference was observed in SUMOylation (Supplementary Figure 5). This could be either due to an effect of CK2 on the SUMOylation machinery as mentioned by the reviewer, or could be due to additional phosphorylation of APC4 at adjacent residues including serines 757, 758, 765 and 808. Given these results, we have de-emphasized the role of phosphorylation in the manuscript.

Fig. 4.C The authors argue that mutating S777 and S779 to alanine resulted in decreased SUMOylation at K772. From the WB it is difficult to judge if the ratio of unmodified to modified APC4 is indeed affected by the S777/779A mutation. Why is there no difference in the level of SUMOylation of wt APC4 compared to the phosphomimic mutant? Is casein kinase II believed to be constitutively active?

In reply: the reviewer correctly mentions that casein kinase II is believed to be constitutively active. This could explain that there is no difference in the level of SUMOylation of wt APC4 compared to the phosphomimic mutant. The ratio of SUMOylated APC4 versus non-SUMOylated APC4 is clearly going down for the S777/779A mutant.

Fig. 4.D Importantly: Why does the treatment with quinalizarin equally affect the mono- and di-SUMOylation. According to the authors, only the SUMOylation of K772 should be affected.

In reply: phosphorylation close to SUMOylation sites has been shown to enhance Ubc9 binding (Mohideen et al. 2009 Nature Structural & Molecular Biology 16, 945 - 952). Since both SUMOylation sites are in close vicinity, we speculate that the SUMOylation of both sites could be carried out in a coordinated fashion. Enhanced Ubc9 binding then would co-regulate SUMOylation of both residues.

Fig. 5.B and C. According to the WB, a substantial fraction of APC4 is unmodified after in vitro

SUMOylation. Yet, according to the silverstain gel the unmodified version of APC4 completely disappears. What happens to APC2 upon in vitro SUMOylation?

In reply: We have now added a less saturated exposure of APC4 to new Figure 6C to demonstrate the large reduction in unmodified APC4 upon SUMOylation. Furthermore, we have probed for APC2 in new Figure 6D to verify equal levels of the protein.

Fig. 5.D Importantly: The authors argue that equal levels of the APC were present during the reactions. However, APC11 WB in the – SUMO reaction clearly indicates uneven levels with the lowest amount being present in the later time points.

In reply: Visualizing APC11 on blot is generally challenging because of its limited size. Whereas we observed the highest activity in the last lane of new Figure 6D, the APC11 levels are not increased. We have now also probed for APC2 in new Figure 6D to verify equal levels of the protein.

In the discussion, the authors discuss how altered kinetics in the destruction of distinct APC substrates could account for the observed phenotypes. This reviewer is puzzled by the provided explanations. How should e.g. altered kinetics of Kif18B destruction account for the observed delay in anaphase onset? Anaphase onset is triggered by silencing the mitotic checkpoint resulting in APC activation. So how could the impaired destruction of Kif18B in SAE1/SAE2 depleted cells delay anaphase onset if one has to assume that destruction of Kif18B does not take place until the mitotic checkpoint is silenced? Do the authors propose that Kif18B is degraded under active checkpoint conditions?

In reply: we agree with the reviewer that the picture is most likely much more complex than the regulation of KIF18B only. Given the large subset of APC/C substrates that contain SIMs, the overall modulation of the interaction of APC/C with these SIM-containing substrates by APC4 SUMOylation might explain the observed delay in metaphase to anaphase progression. We have changed the discussion accordingly.

Reviewer #1 (Remarks to the Author):

The authors have only partially addressed the three comments.

They find:

1. That the SIM site identified in Kif18 that binds SUMO is not required for Kif18B SUMOylation.
2. They were unable to determine if Kif18B ubiquitination is dependent on APC/C SUMOylation *in vivo*.
3. They identified other APC/C substrates harboring SIM sites, but they were unable to determine if ubiquitination of these substrates are dependent on APC/C SUMOylation.

Thus the authors addressed one out of three comments. Comment 3 is not so important, but it is a shame that comment 2 was could not be addressed fully.

Despite this, the manuscript is worthy of publication.

Reviewer #2 (Remarks to the Author):

In the revised version of the manuscript the authors have answered most of the questions and criticism I raised to my satisfaction. Particularly important was an experiment providing *in vivo* evidence that sumoylation of APC4 plays a role in mitotic progression in cells. In the revised version the authors present a replacement experiment, which demonstrates that wt but not a non-sumoylatable version of APC4 can rescue the mitotic delay observed upon depletion of APC4. This experiment together with some other adjustments has substantially improved the manuscript.

There is one issue remaining in regard to Fig. 5. In part B of Fig. 5 the authors show that sumoylation of APC4 C-terminal fragment *in vitro* is increased upon incubation with CK2. In my previous review I requested to include a non-phosphorylatable APC4 mutant as control in this experiment. The authors have added this control (Suppl. Fig. 5) showing however that the mutant version of APC4 is as nicely sumoylated as the wt protein in the presene of CK2. Nevertheless they claim based on Fig. 5B that phosphorylation of the APC4 C-terminal fragment enhances its sumoylation. This conclusion

cannot be drawn from this experiment; all it shows is that the presence of CK2 enhances sumoylation of APC4 in vitro. This could be due to some other potentially phosphorylated sites in APC4 as the authors discuss but could equally well arise from phosphorylation of any component of the sumoylation machinery or even just presence of the CK2 protein independent of its activity. Therefore, the conclusion from this experiment (ll. 286 – 287) needs to be revised accordingly.

Besides this small issue, which can be easily solved, I think the revised manuscript is ready for publication.

Reviewer #3 (Remarks to the Author):

Eifler et al investigate the consequences of APC4 sumoylation. The authors identify two sumoylation sites in the C-terminus of APC4. Phosphorylation of close by sites by CK seems to stimulate APC4 sumoylation. Subsequent studies suggest that sumoylated APC4 has higher ligase activity towards certain substrates, e.g. Kif18B.

Unfortunately, I am still not convinced by the quality of the data and conclusions drawn from the results. Based on the initial ms, the reviewers suggested key experiments that were fundamental to support the main findings of the study. One of those experiments was to test if destruction of Kif18B - but not of securin - is affected in SAE1 and/or SAE2 RNAi cells. Unfortunately, the authors did not perform this experiment and argues why it was not possible. This reviewer is puzzled by the authors' explanation. A simple time-course experiment in Ctrl and SAE1/SAE2-RNAi cells should be possible. Kif18B antibodies are available and the authors actually use them (Fig. 8). However, as outlined below there is an antibody issue with Kif18B and APC4, see below.

The authors still argue that phosphorylation of 772 and 779 might be important for efficient sumoylation of APC4. However, non-phosphorylatable mutants do not support this idea and this reviewer is therefore surprised that the authors still discuss this option.

Fig. 1 still lacks the rescue condition. This is particular important given that the inhibition of protein sumoylation does not correlate with the strength of the mitotic phenotype. Rescue of RNAi should be standard.

Fig. 3, 5 etc. Why does the APC4 band corresponding to the non-sumoylated species of APC4 (input, lane 1) appear in the HIS-SUMO2 pulldown? I guess the pulldown was performed under denaturing conditions. This is apparent in many figures including the supplemental figures and raises concerns regarding the assay conditions.

Fig. 3: According to Fig. 2C, the authors detect APC4 at around 64kDa. Surprisingly, GFP-tagged APC4 runs at exactly the same height, i.e. around 64kDa. How is this possible?

Fig. 4: The mitotic delay upon APC4 depletion is rather mild and therefore it is difficult to judge the rescue efficiency of WT and mutant APC4.

Fig. 4B: A control WB showing equal loading of the protein samples is missing.

Fig. 5D: The authors should include the SAE1/2 depletion control. This would be important. Given that quinalizarin inhibits Ck2 with a K_i value of approximately 50nM, this reviewer wonders how specific an experiment is where 40uM are used. Specifically, if no effect is visible at 20uM, well above the published K_i value.

Fig. 6. The authors suggest "...that SUMOylation of APC4 on the beads was highly efficient. This was further confirmed by immunoblot analysis (Figure 6C)." The immunoblot analysis actually shows that not all APC4 is modified. Given that silver gel staining is more sensitive than immunoblot analysis, this reviewer wonders how the authors explain the fact that "...the unmodified form of APC4 was hardly visible on the silverstain....".

Fig. 8: Importantly, previous studies by the same lab detected Kif18B as a band running between 100 and 150 kDa. According to Fig. 8A, Kif18B runs shortly above 64 kDa. This raises again concerns about the quality of the applied antibodies.

Reviewers' comments:

Reviewer #1 (Remarks to the Author):

The authors have only partially addressed the three comments. They find:

1. That the SIM site identified in Kif18 that binds SUMO is not required for Kif18B SUMOylation.

We suppose that the reviewer meant KIF18B ubiquitylation instead of SUMOylation. Using our peptide approach, we mapped a functional SIM in KIF18B. Mutating this SIM did indeed not reduce the ubiquitylation of KIF18B in vitro. However, we can't rule out that KIF18B contains another SUMO binding site.

2. They were unable to determine if Kif18B ubiquitination is dependent on APC/C SUMOylation in vivo.

We have invested a significant amount of time to study this point of the reviewer but have encountered a set of practical challenges as detailed below. The ubiquitylation of endogenous KIF18B at lysines 62, 304, 688, 783, 795 and 796 is reported in the phosphosite plus post translational modification database (<http://www.phosphosite.org>). Whereas, it is thus clear that KIF18B is a *bona fide* ubiquitin target at the endogenous level in cells, current technical limitations have hampered our efforts to address KIF 18B ubiquitylation in cells:

-We have purified His10-ubiquitin conjugates from cells upon MG132 treatment, nocodazole treatment and release, Thymidine block and release, and subsequently probed for KIF18B, but have been unable to detect ubiquitylation of endogenous KIF18B using the available KIF18B antibody.

-Additionally, we have tried to study changes in overall KIF18B levels upon SUMO E1 knockdown, or in cells expressing APC4 2KR compared to w.t. APC4. We have used the same anti KIF18B antibody, but have not observed changes in any band staining with this antibody. The antibody pattern on total lysates is more complex than expected, impeding our efforts.

-Subsequently, we have overexpressed Flag-KIF18B, performed Flag IPs and blotted for ubiquitin. This resulted in a complex smear, which is missing from the negative control, consistent with KIF18B ubiquitylation. However, since the smear is extending far below the expected size of KIF18B, possibly due to overexpression, this approach appears to be not useful either.

-Thereafter, we have tagged KIF18B at its C-terminus either using a Flag tag or a His10-tag. Cells were transfected with these constructs and purifications were performed, followed by immunoblotting.

Disappointingly, no specific signals could be picked up over the negative control. The reason for the lack of specific signal is not clear. The constructs were verified by sequencing and confirmed to be correct.

3. They identified other APC/C substrates harboring SIM sites, but they were unable to determine if ubiquitination of these substrates are dependent on APC/C SUMOylation.

We have studied whether other APC/C substrates contain consensus SUMO Interaction Motifs. In total, we found 77 potential APC/C substrates in the literature. In this set, we found 35 proteins that contained one or more SUMO Interaction Motifs. The details are shown in the Supplementary table 4 and are as follows:

Substrate	UniProt ID	SUMO interaction motifs
Anillin ¹	Q9NQW6	DPKVEQK IEVIRE IEMSVD (544 – 547) DAANYYY LII LKAGAENMV (833 – 836) ARRNT FELITV RPQREDDR (1058 – 1061) MQKLNQV LVDI RLWQPDAC (1106 – 1109)
Aurora A ²	O14965	None found
Aurora B ³	Q96GD4	None found
B99 ⁴	Q9NYZ3	DDPKKED ILL LAD EKFDFD (9 – 12) NRKTDSR LVDV SPDRG SPP (571 – 575) VGSESR LIDL MTNTPDMN (670 – 673)
BARD1 ⁵	Q99728	GQRRDG PLVLI GSGLSSEQ (570 – 573) DCVMS FELLPLDS ***** (772 – 776)
Bub1 ⁶	O43683	ESNMERR VITISK SEYSVH (211 – 215) IRALRN LIVLL LECKRSR (1073 – 1077)
Cdc6 ⁷	Q99741	TA EKGPMIVLVL DEMDQLD (279 – 283) LSNSHL VLI GIANTLDLTD (313 – 316) LDVCRRA IEIVE SDVKSQT (398 – 402)
Cdc20 ⁸	Q12834	None found
Cdh1 ⁹	Q9UM11	ERRLLRQ IVI QNENTMPRV (13 – 15) TKVKWES VSVL NLFTRIR* (486 – 489)
CDR2 ¹⁰	Q01850	None found
CENPF ¹¹	P49454	None found

Centrin ¹²	O15182	None found
Cdc25A ¹³	P30304	VPTDGKR VIVV FHCFESSE (425 - 428)
CDT1 ¹⁴	Q9H211	None found
CKAP2 ¹⁵	Q8WWK9	ITSPIEN IIAI YEKAILAG (480 - 483) IEEMRHT IVDIL TMKSQEK (502 - 506)
CKS1 ¹⁶	P61024	None found
Claspin ¹⁷	Q9HAW4	None found
Cyclin A2 ¹⁸	P20248	None found
Cyclin B1 ¹⁹	P14635	PLEKVPML LVPV PVSEPVPE (89 - 92) EKLSPEP ILVDT ASPSPE (120 - 124)
Cyclin B3 ²⁰	Q8WWL7	NIMEKPL ILDIST TSKTPN (136 - 140) CDCEAQGL LVL ***** (1393 - 1395)
DRP1 ²¹	O00429	DIIQLPQ IVVVG TQSSGKS (28 - 31) FPKLHDA IVEVV TCLLRKR (464 - 468)
E2C ²²	O00762	None found
E2F1 ²³	Q01094	RLLDSSQ IVIIS AAQDASA (33 - 37) LSHSADG VVDL NWAAEVLK (150 - 153)
E2F3 ²⁴	O00716	SGLKDQT VIVV KAPPETRL (315 - 318)
ECT2 ²⁵	Q9H8V3	None found
FoxM1 ²⁶	Q08050	None found
G9A ²⁷	Q96KQ7	WAAEHKH IEVI RMLLTRGA (798 - 801)
Geminin ²⁸	O75496	None found
GLP1 ²⁷	Q9H9B1	None found
Glutaminase 1 ²⁹	O94925	None found
HEC1 ³⁰	O14777	None found
Hmmr ⁵	O75330	KQSLEEN IVILS KQVEDLN (279 - 283)
HSF2 ³¹	Q03933	NMYGFRK VVHI DGIVKQE (73 - 77) RERISDD IIID VTDDNAD (246 - 250) NCSQYPD IVIVE DDNEDEY (284 - 288)
HURP ⁵	Q15398	PTLEGRI LVEL DETSQGLV (60 - 64)
ID2 ³²	Q02363	LQHVIDY ILDL QIALDSHP (72 - 75) ALDSHP IVSL HHRPGQN (85 - 88)
JNK1 ³³	P45983	None found
JNK2 ³³	P45984	None found

KID ³⁴	Q14807	ESFLKANILGLAAGQRCGA (653 - 656)
KIFC1 ³⁵	Q9BW19	None found
KIF18A ³⁵	Q8NI77	None found
KIF2C ³⁵	Q99661	RMHGKFSLVDLAGNERGAD (490 - 493)
KIF4A ³⁵	O95239	LEIYNEEILDLLCPSREKA (148 - 152) DQELKENVEIICNLQQLIT (463 - 466)
MCL1 ³⁶	Q07820	DELYRQSLEIISRYLREQA (179 - 183)
MFN1 ³⁷	Q8IWA4	CALLRDDLVLVDSPGTDVT (174 - 178) TSRTSMGIIIVGGVIWKTI (599 - 602)
MOAP-1 ³⁸	Q96BY2	None found
MPS1 ³⁹	P33981	None found
NEK2A ⁴⁰	P51955	None found
NIPA ⁴¹	Q86WB0	None found
NLP ⁴²	Q9Y2I6	HDAQRKEIEVLKKDKEKAC (1121 - 1124)
NuSAP ⁵	Q9BXS6	VLGMRRGLILAE***** (436 - 440)
OCT1 ⁴³	P14859	None found
OPA1 ³⁷	O60313	LIDMYSEVLDVLSDYDASY (271 - 275) TQDHLPRVVVVGDSAGKT (291 - 294)
p21 ⁴⁴	P38936	None found
p63 ⁴⁵	Q9H3D4	GMNRRPILIIIVTLETRDQ (322 - 326)
p190RhoGAP ⁴⁶	Q9NRY4	LAKTKKPVVVLTCKDEGV (195 - 199) SDPNIDRINLVILGKDGLA (598 - 602) FLCEVQDIIPIQLVALTDG (880 - 883) HDSTQGKIIITIRNINKAQS (1116 - 1119) ATRTYQTIIEELFIQQCPFF (1424 - 1427)
PAF15 ⁴⁷	Q15004	None found
PFKFB3 ⁴⁸	Q16875	LTNSPTVIVMVGLPARGKT (38 - 41) RDLSLIKVIDVGRRFLVNR (215 - 218)
PIF1 ⁴⁹	Q9H611	None found
PLK1 ⁵⁰	P53350	None found
RacGAP1 ⁵¹	Q9H0H5	TSPMIPSIIVVHCVNEIEQR (365 - 368)
RAP80 ⁵²	Q96RL1	None found
RASSF1A ⁵³	Q9NS23	None found
RCS1 ⁵⁴	Q9BSJ6	None found

Securin ⁵⁵	O95997	None found
SGO1 ⁵⁶	Q5FBB7	None found
SKP2 ¹⁶	Q13309	None found
SnoN ⁵⁷	P12757	None found
Sororin ⁵⁸	Q96FF9	EAAEQFDLLVE***** (249 - 252)
Sp100 ⁵⁹	P23497	HHNQASDIIVISSEDESGS (323 - 327)
TFAM ³⁷	Q00059	None found
TK1 ⁶⁰	P04183	None found
TMPK ⁶¹	P23919	MAARRGALIVLEGVDRAGK (8 - 12)
TOME-1 ⁶²	Q99618	None found
TRB3 ⁶³	Q96RU7	EEEGDREVVLVYG***** (354 - 357)
TPX2 ⁶⁴	Q9ULW0	None found
USP1 ⁶⁵	O94782	None found
USP37 ⁶⁶	Q86T82	EEELLAAVLEISKRDASPS (712 - 716)

We could show that ubiquitylation of the model substrate Hsl1 and ubiquitylation of KIF18B are dependent on SUMOylation of APC4. Time constraints did not permit further testing of additional substrates.

Thus the authors addressed one out of three comments. Comment 3 is not so important, but it is a shame that comment 2 was could not be addressed fully.

Despite this, the manuscript is worthy of publication.

We thank the reviewer for the support.

Reviewer #2 (Remarks to the Author):

In the revised version of the manuscript the authors have answered most of the questions and criticism I raised to my satisfaction. Particularly important was an experiment providing in vivo evidence that sumoylation of APC4 plays a role in mitotic progression in cells. In the revised version the authors present a replacement experiment, which demonstrates that wt but not a non-sumoylatable version of APC4 can rescue the mitotic delay observed upon depletion of APC4. This experiment together with some other adjustments has substantially improved the manuscript.

We are glad to read that the reviewer is satisfied with the revised version of our manuscript, highlighting the in vivo experiment to demonstrate that sumoylation of APC4 is involved in mitotic progression in cells.

There is one issue remaining in regard to Fig. 5. In part B of Fig. 5 the authors show that sumoylation of APC4 C-terminal fragment in vitro is increased upon incubation with CK2. In my previous review I requested to include a non-phosphorylatable APC4 mutant as control in this experiment. The authors have added this control (Suppl. Fig. 5) showing however that the mutant version of APC4 is as nicely sumoylated as the wt protein in the presence of CK2. Nevertheless they claim based on Fig. 5B that phosphorylation of the APC4 C-terminal fragment enhances its sumoylation. This conclusion cannot be drawn from this experiment; all it shows is that the presence of CK2 enhances sumoylation of APC4 in vitro. This could be due to some other potentially phosphorylated sites in APC4 as the authors discuss but could equally well arise from phosphorylation of any component of the sumoylation machinery or even just presence of the CK2 protein independent of its activity. Therefore, the conclusion from this experiment (ll. 286 – 287) needs to be revised accordingly.

We have revised the conclusion from this experiment in the manuscript.

Besides this small issue, which can be easily solved, I think the revised manuscript is ready for publication.

We thank the reviewer for the support.

Reviewer #3 (Remarks to the Author):

Eifler et al investigate the consequences of APC4 sumoylation. The authors identify two sumoylation sites in the C-terminus of APC4. Phosphorylation of close by sites by CK seems to stimulate APC4 sumoylation. Subsequent studies suggest that sumoylated APC4 has higher ligase activity towards certain substrates, e.g. Kif18B.

Unfortunately, I am still not convinced by the quality of the data and conclusions drawn from the results. Based on the initial ms, the reviewers suggested key experiments that were fundamental to support the main findings of the study. One of those experiments was to test if destruction of Kif18B - but not of securin - is affected in SAE1 and/or SAE2 RNAi cells. Unfortunately, the authors did not perform this experiment and argues why it was not possible. This reviewer is puzzled by the authors' explanation. A simple time-course experiment in Ctrl and SAE1/SAE2-RNAi cells should be possible. Kif18B antibodies are available and the authors actually use them (Fig. 8). However, as outlined below there is an antibody issue with Kif18B and APC4, see below.

We agree with the reviewer that the Kif18B antibody is suboptimal as described above, hampering our efforts to address this issue.

The authors still argue that phosphorylation of 772 and 779 might be important for efficient sumoylation of APC4. However, non-phosphorylatable mutants do not support this idea and this reviewer is therefore surprised that the authors still discuss this option.

Mutagenesis of serines 772 and 779 in APC4 caused a decrease in SUMOylation of APC4 in cells (Figure 5c), demonstrating the relevance of these two residues. Nevertheless, we have made textual changes, discussing alternative scenarios explaining our findings as requested by the reviewer.

Fig. 1 still lacks the rescue condition. This is particular important given that the inhibition of protein sumoylation does not correlate with the strength of the mitotic phenotype. Rescue of RNAi should be standard.

Given that we observe significant effects on metaphase to anaphase transition for all three independent knockdown constructs, we consider it highly unlikely that these are off target effects.

Fig. 3, 5 etc. Why does the APC4 band corresponding to the non-sumoylated species of APC4 (input, lane 1) appear in the HIS-SUMO2 pulldown? I guess the pulldown was performed under denaturing

conditions. This is apparent in many figures including the supplemental figures and raises concerns regarding the assay conditions.

This use of His-SUMO and His-ubiquitin is widespread in the field and is highly efficient in purifying modified substrates. Since Ni-NTA beads are relatively sticky a small portion of unmodified APC4 remains on the beads after stringent washing under denaturing conditions. This is not uncommon, and is addressed by including negative controls lacking His-SUMO. The size-shift of SUMOylated APC4 enables straightforward data analysis.

Fig. 3: According to Fig. 2C, the authors detect APC4 at around 64kDa. Surprisingly, GFP-tagged APC4 runs at exactly the same height, i.e. around 64kDa. How is this possible?

We apologize for the labelling mistake. For this protein gel, we used Tris-glycine buffer instead of MOPS buffer, which alters the running behaviour. We thank the reviewer for pointing out this mistake and have corrected the labelling of the blot.

Fig. 4: The mitotic delay upon APC4 depletion is rather mild and therefore it is difficult to judge the rescue efficiency of WT and mutant APC4.

The effect of APC4 knockdown on mitotic delay has been previously shown to be not as efficient as inhibiting the APC/C (Pedersen et al., Nature Communications, 2016). However, we could detect a significant delay and addition of the APC4 WT construct gives a significant rescue while the rescue with the mutant construct was not significant.

Fig. 4B: A control WB showing equal loading of the protein samples is missing.

A Ponceau S staining showing equal loading of the entire lysate is present.

Fig. 5D: The authors should include the SAE1/2 depletion control. This would be important. Given that quinalizarin inhibits Ck2 with a K_i value of approximately 50nM, this reviewer wonders how specific an experiment is where 40uM are used. Specifically, if no effect is visible at 20uM, well above the published K_i value.

We agree that we cannot exclude that this treatment might have other effects that can influence SUMOylation of APC4. We have therefore toned down our conclusion in the main text.

Fig. 6. The authors suggest "...that SUMOylation of APC4 on the beads was highly efficient. This was further confirmed by immunoblot analysis (Figure 6C)." The immunoblot analysis actually shows that not all APC4 is modified. Given that silver gel staining is more sensitive than immunoblot analysis, this reviewer wonders how the authors explain the fact that "...the unmodified form of APC4 was hardly visible on the silverstain....".

The signal intensity for the immunoblot shown in Figure 6C is considerably higher (over saturated signal) compared to the signal intensity for the silverstain depicted in Figure 6B (linear signal), explaining why the non-modified APC4 is visible in Figure 6C. The low signal intensity of non-modified APC4 in the right lane of Figure 6B underlines the efficient SUMOylation of APC4 in vitro.

Fig. 8: Importantly, previous studies by the same lab detected Kif18B as a band running between 100 and 150 kDa. According to Fig. 8A, Kif18B runs shortly above 64 kDa. This raises again concerns about the quality of the applied antibodies.

We agree with the reviewer that the quality of the KIF18B antibody is not ideal. In addition, different gel running and buffer conditions might be responsible for the observed differences in size. We have used the same antibody, but different protein gel running systems in the current study compared to the published study (Tanenbaum et al 2011). In our current study, we are using MOPS buffer for KIF18B experiments instead of TRIS-Glycine buffer as used by Tanenbaum et al. This causes a different running behaviour for the proteins involved. We used the same antibody as previously published.

Reviewer #3 (Remarks to the Author):

Eifler et al investigate the consequences of APC4 sumoylation. The authors identify two sumoylation sites in the C-terminus of APC4. Phosphorylation of close by sites by CK seems to stimulate APC4 sumoylation. Subsequent studies suggest that sumoylated APC4 has higher ligase activity towards certain substrates, e.g. Kif18B.

Unfortunately, I am still not convinced by the quality of the data and conclusions drawn from the results. Based on the initial ms, the reviewers suggested key experiments that were fundamental to support the main findings of the study. One of those experiments was to test if destruction of Kif18B - but not of securin - is affected in SAE1 and/or SAE2 RNAi cells. Unfortunately, the authors did not perform this experiment and argues why it was not possible. This reviewer is puzzled by the authors' explanation. A simple time-course experiment in Ctrl and SAE1/SAE2-RNAi cells should be possible. Kif18B antibodies are available and the authors actually use them (Fig. 8). However, as outlined below there is an antibody issue with Kif18B and APC4, see below.

We agree with the reviewer that the Kif18B antibody is suboptimal as described above, hampering our efforts to address this issue.

This issue was originally raised by Reviewer 1 and not by Reviewer 3. As Reviewer 1 commented, I agree that it is a pity that this point could not be solved but still the study is worthy to be published. The Kif18B antibody issue however is one that needs to be addressed (see further below).

The authors still argue that phosphorylation of 772 and 779 might be important for efficient sumoylation of APC4. However, non-phosphorylatable mutants do not support this idea and this reviewer is therefore surprised that the authors still discuss this option.

Mutagenesis of serines 772 and 779 in APC4 caused a decrease in SUMOylation of APC4 in cells (Figure 5c), demonstrating the relevance of these two residues. Nevertheless, we have made textual changes, discussing alternative scenarios explaining our findings as requested by the reviewer.

I will comment on this further below.

Fig. 1 still lacks the rescue condition. This is particular important given that the inhibition of protein sumoylation does not correlate with the strength of the mitotic phenotype. Rescue of RNAi should be standard.

Given that we observe significant effects on metaphase to anaphase transition for all three independent knockdown constructs, we consider it highly unlikely that these are off target effects.

The fact that sumoylation is important for mitosis and for the meta- to anaphase transition is known and accepted. From this perspective, I consider these initial siRNA experiments rather as story-telling experiments, which I anyways expect to result in rather pleiotropic effects. Therefore I think one can accept the authors' argument that three independent siRNAs result in a similar phenotype even without a rescue experiment.

Fig. 3, 5 etc. Why does the APC4 band corresponding to the non-sumoylated species of APC4 (input, lane 1) appear in the HIS-SUMO2 pulldown? I guess the pulldown was performed under denaturing conditions. This is apparent in many figures including the supplemental figures and raises concerns regarding the assay conditions.

This use of His-SUMO and His-ubiquitin is widespread in the field and is highly efficient in purifying modified substrates. Since Ni-NTA beads are relatively sticky a small portion of unmodified APC4 remains on the beads after stringent washing under denaturing conditions. This is not uncommon, and is addressed by including negative controls lacking His-SUMO. The size-shift of SUMOylated APC4 enables straightforward data analysis.

I agree with the authors' explanation; it is indeed a common observation that the unmodified form is often coming along in such purifications, likely due to stickiness.

Fig. 3: According to Fig. 2C, the authors detect APC4 at around 64kDa. Surprisingly, GFP-tagged APC4 runs at exactly the same height, i.e. around 64kDa. How is this possible?

We apologize for the labelling mistake. For this protein gel, we used Tris-glycine buffer instead of MOPS buffer, which alters the running behaviour. We thank the reviewer for pointing out this mistake and have corrected the labelling of the blot.

Problem solved

Fig. 4: The mitotic delay upon APC4 depletion is rather mild and therefore it is difficult to judge the rescue efficiency of WT and mutant APC4.

The effect of APC4 knockdown on mitotic delay has been previously shown to be not as efficient as inhibiting the APC/C (Pedersen et al., Nature Communications, 2016). However, we could detect a significant delay and addition of the APC4 WT construct gives a significant rescue while the rescue with the mutant construct was not significant.

From my perspective I consider these experiments as sufficiently convincing; this I already stated in my own reply to the revised version of this manuscript.

Fig. 4B: A control WB showing equal loading of the protein samples is missing.

A Ponceau S staining showing equal loading of the entire lysate is present.

The Ponceau staining is sufficient in this case to control for equal loading.

Fig. 5D: The authors should include the SAE1/2 depletion control. This would be important. Given that quinalizarin inhibits Ck2 with a K_i value of approximately 50nM, this reviewer wonders how specific an experiment is where 40uM are used. Specifically, if no effect is visible at 20uM, well above the published K_i value.

We agree that we cannot exclude that this treatment might have other effects that can influence SUMOylation of APC4. We have therefore toned down our conclusion in the main text.

Reviewer 1's point is well taken. I have to admit that I was not aware that CK2 has such a low K_i , and have so far overlooked this aspect. Taking this information into account the most likely conclusion from this experiment is that phosphorylation by CK2 is unlikely to have an effect on APC4 sumoylation. In addition, the authors are also not able to observe any effect of phosphorylation on S777 and S779 by CK2 in vitro. In this context, I would like to point out that I noticed that the authors did not really revise the manuscript according to my comment: ll. 286-287 are still unchanged (this has to be corrected or removed!!). The respective experiment (Fig. 5B) only allows concluding that the addition of CK2 into the sumoylation reaction increases APC4 sumoylation, and this is not due to phosphorylation of S777/S779 by CK2 (Suppl. Fig 5).

Taking everything together, I therefore suggest that you could ask the authors to remove the CK2 data (in vitro as well as in vivo) from the manuscript as they do not support what the authors try to say. On the one hand, this would suffice some of Reviewer 3's concerns; on the other hand, it is also not essential for the story as the potential regulation by phosphorylation is only a sideline anyways.

Fig. 6. The authors suggest "...that SUMOylation of APC4 on the beads was highly efficient. This was further confirmed by immunoblot analysis (Figure 6C)." The immunoblot analysis actually shows that not all APC4 is modified. Given that silver gel staining is more sensitive than immunoblot analysis, this

reviewer wonders how the authors explain the fact that “...the unmodified form of APC4 was hardly visible on the silverstain...”.

The signal intensity for the immunoblot shown in Figure 6C is considerably higher (over saturated signal) compared to the signal intensity for the silverstain depicted in Figure 6B (linear signal), explaining why the non-modified APC4 is visible in Figure 6C. The low signal intensity of non-modified APC4 in the right lane of Figure 6B underlines the efficient SUMOylation of APC4 in vitro.

Generally, I consider this explanation a valid argument. I still think the authors should check their molecular weight marker; in Fig. 6B, unmodified APC4 runs between 64 and 97 kDa, in Fig. 6C just above the 64kDa marker. Something must be slightly off here. Having a consistent and correct labeling would help to avoid confusions and doubts.

Fig. 8: Importantly, previous studies by the same lab detected Kif18B as a band running between 100 and 150 kDa. According to Fig. 8A, Kif18B runs shortly above 64 kDa. This raises again concerns about the quality of the applied antibodies.

We agree with the reviewer that the quality of the KIF18B antibody is not ideal. In addition, different gel running and buffer conditions might be responsible for the observed differences in size. We have used the same antibody, but different protein gel running systems in the current study compared to the published study (Tanenbaum et al 2011). In our current study, we are using MOPS buffer for KIF18B experiments instead of TRIS-Glycine buffer as used by Tanenbaum et al. This causes a different running behaviour for the proteins involved. We used the same antibody as previously published.

The quality of the used Kif18b antibody is a serious and justified point of concern for two reasons. 1. It is quite common that proteins run at a larger apparent molecular weight than expected, however the opposite as observed in this manuscript is rather uncommon. 2. The antibody indeed seems to be of low quality as it recognizes many proteins in immunoblotting (Suppl. Fig. 8A,B,D). How can the authors be sure that the bands they claim to be Kif18b is Kif18b, especially if it is running at an unexpected molecular weight? Even in the recombinant assay (Suppl. Fig.8D) it recognizes two proteins with roughly equal signal intensity. A control sample without substrate or at least a control blot with the Flag antibody (as they are using Flag-Kif18b) is unfortunately missing in this experiment.

The authors even admit that the quality of their antibody is suboptimal; unfortunately however, they only try to argue with potential explanations (different gel running systems) instead of simply proving that the protein band they claim to correspond to Kif18b is indeed Kif18b or instead of using a reliable antibody for their analyses. This is not acceptable and does certainly not suffice to address Reviewer 3's concern. It is also essential for the story to guarantee that we are indeed looking at Kif18b.

A minimal requirement in this case is that the authors at least validate the Kif18b antibody in two ways. 1. The authors need to show that this band at around 64kDa is sensitive to siRNA knockdown against Kif18b in cells. 2. If their recombinant Kif18b construct is a full-length construct (is it? what is the

theoretical molecular weight for the protein encoded by this construct?), they should demonstrate that both, the Kif18b but also the Flag antibody recognize the same protein band at around 64 kDa.

Reviewer #3 (Remarks to the Author):

Eifler et al investigate the consequences of APC4 sumoylation. The authors identify two sumoylation sites in the C-terminus of APC4. Phosphorylation of close by sites by CK seems to stimulate APC4 sumoylation. Subsequent studies suggest that sumoylated APC4 has higher ligase activity towards certain substrates, e.g. Kif18B.

Unfortunately, I am still not convinced by the quality of the data and conclusions drawn from the results. Based on the initial ms, the reviewers suggested key experiments that were fundamental to support the main findings of the study. One of those experiments was to test if destruction of Kif18B - but not of securin - is affected in SAE1 and/or SAE2 RNAi cells. Unfortunately, the authors did not perform this experiment and argues why it was not possible. This reviewer is puzzled by the authors' explanation. A simple time-course experiment in Ctrl and SAE1/SAE2-RNAi cells should be possible. Kif18B antibodies are available and the authors actually use them (Fig. 8). However, as outlined below there is an antibody issue with Kif18B and APC4, see below.

We agree with the reviewer that the Kif18B antibody is suboptimal as described above, hampering our efforts to address this issue.

This issue was originally raised by Reviewer 1 and not by Reviewer 3. As Reviewer 1 commented, I agree that it is a pity that this point could not be solved but still the study is worthy to be published. The Kif18B antibody issue however is one that needs to be addressed (see further below).

In reply: We appreciate the support. The Kif18B antibody issue has been addressed below.

The authors still argue that phosphorylation of 772 and 779 might be important for efficient sumoylation of APC4. However, non-phosphorylatable mutants do not support this idea and this reviewer is therefore surprised that the authors still discuss this option.

Mutagenesis of serines 772 and 779 in APC4 caused a decrease in SUMOylation of APC4 in cells (Figure 5c), demonstrating the relevance of these two residues. Nevertheless, we have made textual changes, discussing alternative scenarios explaining our findings as requested by the reviewer.

I will comment on this further below.

Fig. 1 still lacks the rescue condition. This is particular important given that the inhibition of protein sumoylation does not correlate with the strength of the mitotic phenotype. Rescue of RNAi should be standard.

Given that we observe significant effects on metaphase to anaphase transition for all three independent knockdown constructs, we consider it highly unlikely that these are off target effects.

The fact that sumoylation is important for mitosis and for the meta- to anaphase transition is known and accepted. From this perspective, I consider these initial siRNA experiments rather as story-telling experiments, which I anyways expect to result in rather pleiotropic effects. Therefore I think one can

accept the authors' argument that three independent siRNAs result in a similar phenotype even without a rescue experiment.

In reply: We appreciate the support.

Fig. 3, 5 etc. Why does the APC4 band corresponding to the non-sumoylated species of APC4 (input, lane 1) appear in the HIS-SUMO2 pulldown? I guess the pulldown was performed under denaturing conditions. This is apparent in many figures including the supplemental figures and raises concerns regarding the assay conditions.

This use of His-SUMO and His-ubiquitin is widespread in the field and is highly efficient in purifying modified substrates. Since Ni-NTA beads are relatively sticky a small portion of unmodified APC4 remains on the beads after stringent washing under denaturing conditions. This is not uncommon, and is addressed by including negative controls lacking His-SUMO. The size-shift of SUMOylated APC4 enables straightforward data analysis.

I agree with the authors' explanation; it is indeed a common observation that the unmodified form is often coming along in such purifications, likely due to stickiness.

Fig. 3: According to Fig. 2C, the authors detect APC4 at around 64kDa. Surprisingly, GFP-tagged APC4 runs at exactly the same height, i.e. around 64kDa. How is this possible?

We apologize for the labelling mistake. For this protein gel, we used Tris-glycine buffer instead of MOPS buffer, which alters the running behaviour. We thank the reviewer for pointing out this mistake and have corrected the labelling of the blot.

Problem solved

Fig. 4: The mitotic delay upon APC4 depletion is rather mild and therefore it is difficult to judge the rescue efficiency of WT and mutant APC4.

The effect of APC4 knockdown on mitotic delay has been previously shown to be not as efficient as inhibiting the APC/C (Pedersen et al., Nature Communications, 2016). However, we could detect a significant delay and addition of the APC4 WT construct gives a significant rescue while the rescue with the mutant construct was not significant.

From my perspective I consider these experiments as sufficiently convincing; this I already stated in my own reply to the revised version of this manuscript.

In reply: We appreciate the support.

Fig. 4B: A control WB showing equal loading of the protein samples is missing.

A Ponceau S staining showing equal loading of the entire lysate is present.

The Ponceau staining is sufficient in this case to control for equal loading.

In reply: We appreciate the support.

Fig. 5D: The authors should include the SAE1/2 depletion control. This would be important. Given that quinalizarin inhibits Ck2 with a K_i value of approximately 50nM, this reviewer wonders how specific an experiment is where 40uM are used. Specifically, if no effect is visible at 20uM, well above the published K_i value.

We agree that we cannot exclude that this treatment might have other effects that can influence SUMOylation of APC4. We have therefore toned down our conclusion in the main text.

Reviewer 1's point is well taken. I have to admit that I was not aware that CK2 has such a low K_i , and have so far overlooked this aspect. Taking this information into account the most likely conclusion from this experiment is that phosphorylation by CK2 is unlikely to have an effect on APC4 sumoylation. In addition, the authors are also not able to observe any effect of phosphorylation on S777 and S779 by CK2 in vitro. In this context, I would like to point out that I noticed that the authors did not really revise the manuscript according to my comment: ll. 286-287 are still unchanged (this has to be corrected or removed!!). The respective experiment (Fig. 5B) only allows concluding that the addition of CK2 into the sumoylation reaction increases APC4 sumoylation, and this is not due to phosphorylation of S777/S779 by CK2 (Suppl. Fig 5).

Taking everything together, I therefore suggest that you could ask the authors to remove the CK2 data (in vitro as well as in vivo) from the manuscript as they do not support what the authors try to say. On the one hand, this would suffice some of Reviewer 3's concerns; on the other hand, it is also not essential for the story as the potential regulation by phosphorylation is only a sideline anyways.

In reply: We have removed the CK2 data (both in vitro data and data from cellular experiments) from the manuscript. We agree with the reviewer that the potential regulation of APC4 sumoylation by phosphorylation is not important for the main message. The remaining phosphorylation data has therefore been moved to the supplement (Supplementary Figure 5) to avoid distraction.

Fig. 6. The authors suggest "...that SUMOylation of APC4 on the beads was highly efficient. This was further confirmed by immunoblot analysis (Figure 6C)." The immunoblot analysis actually shows that not all APC4 is modified. Given that silver gel staining is more sensitive than immunoblot analysis, this reviewer wonders how the authors explain the fact that "...the unmodified form of APC4 was hardly visible on the silverstain...".

The signal intensity for the immunoblot shown in Figure 6C is considerably higher (over saturated signal) compared to the signal intensity for the silverstain depicted in Figure 6B (linear signal), explaining why the non-modified APC4 is visible in Figure 6C. The low signal intensity of non-modified APC4 in the right

lane of Figure 6B underlines the efficient SUMOylation of APC4 in vitro.

Generally, I consider this explanation a valid argument. I still think the authors should check their molecular weight marker; in Fig. 6B, unmodified APC4 runs between 64 and 97 kDa, in Fig. 6C just above the 64kDa marker. Something must be slightly off here. Having a consistent and correct labeling would help to avoid confusions and doubts.

In reply: We have double checked and corrected the molecular weight markers to ensure consistent and correct labelling.

Fig. 8: Importantly, previous studies by the same lab detected Kif18B as a band running between 100 and 150 kDa. According to Fig. 8A, Kif18B runs shortly above 64 kDa. This raises again concerns about the quality of the applied antibodies.

We agree with the reviewer that the quality of the KIF18B antibody is not ideal. In addition, different gel running and buffer conditions might be responsible for the observed differences in size. We have used the same antibody, but different protein gel running systems in the current study compared to the published study (Tanenbaum et al 2011). In our current study, we are using MOPS buffer for KIF18B experiments instead of TRIS-Glycine buffer as used by Tanenbaum et al. This causes a different running behaviour for the proteins involved. We used the same antibody as previously published.

The quality of the used Kif18b antibody is a serious and justified point of concern for two reasons. 1. It is quite common that proteins run at a larger apparent molecular weight than expected, however the opposite as observed in this manuscript is rather uncommon. 2. The antibody indeed seems to be of low quality as it recognizes many proteins in immunoblotting (Suppl. Fig. 8A,B,D). How can the authors be sure that the bands they claim to be Kif18b is Kif18b, especially if it is running at an unexpected molecular weight? Even in the recombinant assay (Suppl. Fig.8D) it recognizes two proteins with roughly equal signal intensity. A control sample without substrate or at least a control blot with the Flag antibody (as they are using Flag-Kif18b) is unfortunately missing in this experiment.

The authors even admit that the quality of their antibody is suboptimal; unfortunately however, they only try to argue with potential explanations (different gel running systems) instead of simply proving that the protein band they claim to correspond to Kif18b is indeed Kif18b or instead of using a reliable antibody for their analyses. This is not acceptable and does certainly not suffice to address Reviewer 3's concern. It is also essential for the story to guarantee that we are indeed looking at Kif18b.

A minimal requirement in this case is that the authors at least validate the Kif18b antibody in two ways. 1. The authors need to show that this band at around 64kDa is sensitive to siRNA knockdown against Kif18b in cells. 2. If their recombinant Kif18b construct is a full-length construct (is it? what is the theoretical molecular weight for the protein encoded by this construct?), they should demonstrate that both, the Kif18b but also the Flag antibody recognize the same protein band at around 64 kDa.

In reply: We thank the reviewer for the constructive comments.

1. We have carried out the Kif18b siRNA experiment and found that the two dominant bands recognized by the antibody in lysates from HeLa cells are sensitive to siRNA treatment. These bands run slightly

below the 97 kDa marker whereas the predicated size in Uniprot for Kif18b is 94 kDa, so this is consistent. These results are now added to the supplement (Supplementary Figure 7). The size markers for the in vitro experiment with Kif18b have now been carefully verified and corrected and are consistent.

2. The recombinant Kif18b construct is a kind gift from Prof. René Medema (Tanenbaum 2011 *Curr. Biol.* 21:1356-1365) and encodes full-length Kif18b indeed. For our in vitro experiments, we expressed the protein in mammalian cells, prepared lysates, performed immunoprecipitation with Flag antibody and detected the protein with the Kif18b antibody. These results are consistent and demonstrate that both the Flag antibody and the validated Kif18b recognize the protein. The mass spec result on the affinity of sumoylated APC/C for endogenous Kif18b provides antibody-independent support. In summary, we believe that our results on Kif18b are consistent and reliable and hope that our manuscript will now finally be published in Nature Communications.

Reviewer #3 (Remarks to the Author):

Eifler et al investigate the consequences of APC4 sumoylation. The authors identify two sumoylation sites in the C-terminus of APC4. Phosphorylation of close by sites by CK seems to stimulate APC4 sumoylation. Subsequent studies suggest that sumoylated APC4 has higher ligase activity towards certain substrates, e.g. Kif18B.

Unfortunately, I am still not convinced by the quality of the data and conclusions drawn from the results. Based on the initial ms, the reviewers suggested key experiments that were fundamental to support the main findings of the study. One of those experiments was to test if destruction of Kif18B - but not of securin - is affected in SAE1 and/or SAE2 RNAi cells. Unfortunately, the authors did not perform this experiment and argues why it was not possible. This reviewer is puzzled by the authors' explanation. A simple time-course experiment in Ctrl and SAE1/SAE2-RNAi cells should be possible. Kif18B antibodies are available and the authors actually use them (Fig. 8). However, as outlined below there is an antibody issue with Kif18B and APC4, see below.

We agree with the reviewer that the Kif18B antibody is suboptimal as described above, hampering our efforts to address this issue.

This issue was originally raised by Reviewer 1 and not by Reviewer 3. As Reviewer 1 commented, I agree that it is a pity that this point could not be solved but still the study is worthy to be published. The Kif18B antibody issue however is one that needs to be addressed (see further below).

In reply: We appreciate the support. The Kif18B antibody issue has been addressed below.

The authors still argue that phosphorylation of 772 and 779 might be important for efficient sumoylation of APC4. However, non-phosphorylatable mutants do not support this idea and this reviewer is therefore surprised that the authors still discuss this option.

Mutagenesis of serines 772 and 779 in APC4 caused a decrease in SUMOylation of APC4 in cells (Figure 5c), demonstrating the relevance of these two residues. Nevertheless, we have made textual changes, discussing alternative scenarios explaining our findings as requested by the reviewer.

I will comment on this further below.

Fig. 1 still lacks the rescue condition. This is particular important given that the inhibition of protein sumoylation does not correlate with the strength of the mitotic phenotype. Rescue of RNAi should be standard.

Given that we observe significant effects on metaphase to anaphase transition for all three independent knockdown constructs, we consider it highly unlikely that these are off target effects.

The fact that sumoylation is important for mitosis and for the meta- to anaphase transition is known and accepted. From this perspective, I consider these initial siRNA experiments rather as story-telling experiments, which I anyways expect to result in rather pleiotropic effects. Therefore I think one can

accept the authors' argument that three independent siRNAs result in a similar phenotype even without a rescue experiment.

In reply: We appreciate the support.

Fig. 3, 5 etc. Why does the APC4 band corresponding to the non-sumoylated species of APC4 (input, lane 1) appear in the HIS-SUMO2 pulldown? I guess the pulldown was performed under denaturing conditions. This is apparent in many figures including the supplemental figures and raises concerns regarding the assay conditions.

This use of His-SUMO and His-ubiquitin is widespread in the field and is highly efficient in purifying modified substrates. Since Ni-NTA beads are relatively sticky a small portion of unmodified APC4 remains on the beads after stringent washing under denaturing conditions. This is not uncommon, and is addressed by including negative controls lacking His-SUMO. The size-shift of SUMOylated APC4 enables straightforward data analysis.

I agree with the authors' explanation; it is indeed a common observation that the unmodified form is often coming along in such purifications, likely due to stickiness.

Fig. 3: According to Fig. 2C, the authors detect APC4 at around 64kDa. Surprisingly, GFP-tagged APC4 runs at exactly the same height, i.e. around 64kDa. How is this possible?

We apologize for the labelling mistake. For this protein gel, we used Tris-glycine buffer instead of MOPS buffer, which alters the running behaviour. We thank the reviewer for pointing out this mistake and have corrected the labelling of the blot.

Problem solved

Fig. 4: The mitotic delay upon APC4 depletion is rather mild and therefore it is difficult to judge the rescue efficiency of WT and mutant APC4.

The effect of APC4 knockdown on mitotic delay has been previously shown to be not as efficient as inhibiting the APC/C (Pedersen et al., Nature Communications, 2016). However, we could detect a significant delay and addition of the APC4 WT construct gives a significant rescue while the rescue with the mutant construct was not significant.

From my perspective I consider these experiments as sufficiently convincing; this I already stated in my own reply to the revised version of this manuscript.

In reply: We appreciate the support.

Fig. 4B: A control WB showing equal loading of the protein samples is missing.

A Ponceau S staining showing equal loading of the entire lysate is present.

The Ponceau staining is sufficient in this case to control for equal loading.

In reply: We appreciate the support.

Fig. 5D: The authors should include the SAE1/2 depletion control. This would be important. Given that quinalizarin inhibits Ck2 with a K_i value of approximately 50nM, this reviewer wonders how specific an experiment is where 40uM are used. Specifically, if no effect is visible at 20uM, well above the published K_i value.

We agree that we cannot exclude that this treatment might have other effects that can influence SUMOylation of APC4. We have therefore toned down our conclusion in the main text.

Reviewer 1's point is well taken. I have to admit that I was not aware that CK2 has such a low K_i , and have so far overlooked this aspect. Taking this information into account the most likely conclusion from this experiment is that phosphorylation by CK2 is unlikely to have an effect on APC4 sumoylation. In addition, the authors are also not able to observe any effect of phosphorylation on S777 and S779 by CK2 in vitro. In this context, I would like to point out that I noticed that the authors did not really revise the manuscript according to my comment: ll. 286-287 are still unchanged (this has to be corrected or removed!!). The respective experiment (Fig. 5B) only allows concluding that the addition of CK2 into the sumoylation reaction increases APC4 sumoylation, and this is not due to phosphorylation of S777/S779 by CK2 (Suppl. Fig 5).

Taking everything together, I therefore suggest that you could ask the authors to remove the CK2 data (in vitro as well as in vivo) from the manuscript as they do not support what the authors try to say. On the one hand, this would suffice some of Reviewer 3's concerns; on the other hand, it is also not essential for the story as the potential regulation by phosphorylation is only a sideline anyways.

In reply: We have removed the CK2 data (both in vitro data and data from cellular experiments) from the manuscript. We agree with the reviewer that the potential regulation of APC4 sumoylation by phosphorylation is not important for the main message. The remaining phosphorylation data has therefore been moved to the supplement (Supplementary Figure 5) to avoid distraction.

Fig. 6. The authors suggest "...that SUMOylation of APC4 on the beads was highly efficient. This was further confirmed by immunoblot analysis (Figure 6C)." The immunoblot analysis actually shows that not all APC4 is modified. Given that silver gel staining is more sensitive than immunoblot analysis, this reviewer wonders how the authors explain the fact that "...the unmodified form of APC4 was hardly visible on the silverstain...".

The signal intensity for the immunoblot shown in Figure 6C is considerably higher (over saturated signal) compared to the signal intensity for the silverstain depicted in Figure 6B (linear signal), explaining why the non-modified APC4 is visible in Figure 6C. The low signal intensity of non-modified APC4 in the right

lane of Figure 6B underlines the efficient SUMOylation of APC4 in vitro.

Generally, I consider this explanation a valid argument. I still think the authors should check their molecular weight marker; in Fig. 6B, unmodified APC4 runs between 64 and 97 kDa, in Fig. 6C just above the 64kDa marker. Something must be slightly off here. Having a consistent and correct labeling would help to avoid confusions and doubts.

In reply: We have double checked and corrected the molecular weight markers to ensure consistent and correct labelling.

Fig. 8: Importantly, previous studies by the same lab detected Kif18B as a band running between 100 and 150 kDa. According to Fig. 8A, Kif18B runs shortly above 64 kDa. This raises again concerns about the quality of the applied antibodies.

We agree with the reviewer that the quality of the KIF18B antibody is not ideal. In addition, different gel running and buffer conditions might be responsible for the observed differences in size. We have used the same antibody, but different protein gel running systems in the current study compared to the published study (Tanenbaum et al 2011). In our current study, we are using MOPS buffer for KIF18B experiments instead of TRIS-Glycine buffer as used by Tanenbaum et al. This causes a different running behaviour for the proteins involved. We used the same antibody as previously published.

The quality of the used Kif18b antibody is a serious and justified point of concern for two reasons. 1. It is quite common that proteins run at a larger apparent molecular weight than expected, however the opposite as observed in this manuscript is rather uncommon. 2. The antibody indeed seems to be of low quality as it recognizes many proteins in immunoblotting (Suppl. Fig. 8A,B,D). How can the authors be sure that the bands they claim to be Kif18b is Kif18b, especially if it is running at an unexpected molecular weight? Even in the recombinant assay (Suppl. Fig.8D) it recognizes two proteins with roughly equal signal intensity. A control sample without substrate or at least a control blot with the Flag antibody (as they are using Flag-Kif18b) is unfortunately missing in this experiment.

The authors even admit that the quality of their antibody is suboptimal; unfortunately however, they only try to argue with potential explanations (different gel running systems) instead of simply proving that the protein band they claim to correspond to Kif18b is indeed Kif18b or instead of using a reliable antibody for their analyses. This is not acceptable and does certainly not suffice to address Reviewer 3's concern. It is also essential for the story to guarantee that we are indeed looking at Kif18b.

A minimal requirement in this case is that the authors at least validate the Kif18b antibody in two ways. 1. The authors need to show that this band at around 64kDa is sensitive to siRNA knockdown against Kif18b in cells. 2. If their recombinant Kif18b construct is a full-length construct (is it? what is the theoretical molecular weight for the protein encoded by this construct?), they should demonstrate that both, the Kif18b but also the Flag antibody recognize the same protein band at around 64 kDa.

In reply: We thank the reviewer for the constructive comments.

1. We have carried out the Kif18b siRNA experiment and found that the two dominant bands recognized by the antibody in lysates from HeLa cells are sensitive to siRNA treatment. These bands run slightly

below the 97 kDa marker whereas the predicated size in Uniprot for Kif18b is 94 kDa, so this is consistent. These results are now added to the supplement (Supplementary Figure 7). The size markers for the in vitro experiment with Kif18b have now been carefully verified and corrected and are consistent.

2. The recombinant Kif18b construct is a kind gift from Prof. René Medema (Tanenbaum 2011 *Curr. Biol.* 21:1356-1365) and encodes full-length Kif18b indeed. For our in vitro experiments, we expressed the protein in mammalian cells, prepared lysates, performed immunoprecipitation with Flag antibody and detected the protein with the Kif18b antibody. These results are consistent and demonstrate that both the Flag antibody and the validated Kif18b recognize the protein. The mass spec result on the affinity of sumoylated APC/C for endogenous Kif18b provides antibody-independent support. In summary, we believe that our results on Kif18b are consistent and reliable and hope that our manuscript will now finally be published in Nature Communications.